# Learning Strategy-Aware Linear Classifiers

**Yiling Chen**
Harvard University
`yiling@seas.harvard.edu`

**Yang Liu**
UC Santa Cruz
`yangliu@ucsc.edu`

**Chara Podimata**
Harvard University
`podimata@g.harvard.edu`

## Abstract

We address the question of repeatedly learning linear classifiers against agents who are *strategically* trying to *game* the deployed classifiers, and we use the *Stackelberg regret* to measure the performance of our algorithms. First, we show that Stackelberg and external regret for the problem of strategic classification are *strongly incompatible*: i.e., there exist worst-case scenarios, where *any* sequence of actions providing *sublinear* external regret might result in *linear* Stackelberg regret and vice versa. Second, we present a strategy-aware algorithm for minimizing the Stackelberg regret for which we prove nearly matching upper and lower regret bounds. Finally, we provide simulations to complement our theoretical analysis. Our results advance the growing literature of learning from revealed preferences, which has so far focused on "smoother" assumptions from the perspective of the learner and the agents respectively.

## 1 Introduction

As Machine Learning (ML) algorithms become increasingly involved in real-life decision making, the agents that they interact with tend to be neither stochastic nor adversarial. Rather, they are *strategic*. For example, consider a college that wishes to deploy an ML algorithm to make admissions decisions. Student candidates might try to manipulate their test scores in an effort to fool the classifier. Or think about email spammers who are trying to manipulate their emails in an effort to fool the ML classifier and land in the non-spam inboxes. Importantly, in both examples the agents (students and spammers respectively) do not want to sabotage the classification algorithm only for the sake of harming its performance. They merely want to game it for their own benefit. And this is precisely what differentiates them from being fully adversarial.

Motivated by the problem of classifying spam emails, we focus on the problem of learning an unknown *linear* classifier, when the training data come in an *online* fashion from *strategic* agents, who can alter their feature vectors to *game* the classifier. We model the interplay between the learner and the strategic agents[1] as a repeated *Stackelberg game* over $T$ timesteps. In a repeated Stackelberg game, the learner ("leader") *commits* to an action, and then, the agent ("follower") best-responds to it, i.e., reports something that maximizes his underlying utility. The learner's goal is to minimize her *Stackelberg regret*, which is the difference between her cumulative loss and the cumulative loss of her best-fixed action in hindsight, *had she given the agent the opportunity to best-respond to it*.

**Our Contributions.**

- We study a general model of learning interaction in strategic classification settings where the agents' true datapoint remains hidden from the learner, the agents can misreport within a ball of radius $\delta$ of their true datapoint (termed *$\delta$-bounded, myopically rational ($\delta$-BMR) agents*), and the learner measures her performance using the *binary loss*. This model departs significantly from the smooth utility and loss functions used so far for strategic classification (Sec. 2).

- We prove that in strategic classification settings against $\delta$-BMR agents *simultaneously* achieving sublinear external and Stackelberg regret is in general impossible (*strong incompatibility*) (i.e., application of standard no-external regret algorithms might be unhelpful (Sec. 3)).

- Taking advantage of the structure of the responses of $\delta$-BMR agents while working in the *dual* space of the learner, we propose an adaptive discretization algorithm (GRINDER), which uses access to an oracle. GRINDER's novelty is that it assumes no stochasticity for the adaptive discretization (Sec. 4).

- We prove that the regret guarantees of GRINDER remain *unchanged* in order even when the learner is given access to a *noisy* oracle, accommodating more settings in practice (Sec. 4).

- We prove nearly matching lower bounds for strategic classification against $\delta$-BMR agents (Sec. 5).

- We provide simulations implementing GRINDER both for continuous and discrete action spaces, and using both an accurate and an approximation oracle (Sec. 4.1).

**Our Techniques.**

- In order to prove the incompatibility results of the regret notions in strategic classification, we present a formal framework, which may be of independent interest.

- To overcome the non-smooth utility and loss functions, we work on the *dual* space, which provides information about various *regions* of the learner's action space, *despite never observing the agent's true datapoint*. These regions (polytopes) relate to the partitions that GRINDER creates.

- To deal with the learner's action space being *continuous* (i.e., containing infinite actions), we use the fact that *all actions* within a polytope share the same *history* of estimated losses. So, passing information down to a recently partitioned polytope becomes a simple volume reweighting.

- To account for all the actions in the continuous action space, we present a formulation of the standard EXP3 algorithm that takes advantage of the polytope partitioning process.

- For bounding the variance of our polytope-based loss estimator, we develop a polytope-based variant of a well-known graph-theoretic lemma ([1, Lem. 5]), which has been crucial in the analysis of online learning settings with feedback graphs. Such a variant is mandatory, since direct application of [1, Lem. 5] in settings with continuous action spaces yields vacuous[2] regret.

- We develop a generalization of standard techniques for proving regret lower bounds in *strategic* settings, where the datapoints that the agents report *change* in response to the learner's actions.

**Related Work.** Our work is primarily related to the literature on *learning using data from strategic sources* (e.g., [16, 13, 31, 17, 14, 6, 7, 25, 9, 27, 28]). Our work is also related to learning in Stackelberg Security Games (SSGs) ([24, 26, 8]) and especially, the work of Balcan et al. [4], who study information theoretic sublinear Stackelberg regret[3] algorithms for the learner. In SSGs, all utilities are linear, a property not present in strategic classification against $\delta$-BMR agents. Finally, our work is related to the literature in online learning with *partial* (see [11, 33, 23]) and *graph-structured* feedback [1, 15]. Adaptive discretization algorithms were studied for stochastic Lipschitz settings in [22, 10], but in learning against $\delta$-BMR agents, the loss is neither stochastic nor Lipschitz.

## 2 Model and Preliminaries

We use the spam email application as a running example to setup our model. Each agent has an email that is either a spam or a non-spam. Given a classifier, the agent can alter his email to a certain degree in order to bypass the email filter and have his email be classified as non-spam. Such manipulation is costly. Each agent chooses a manipulation to maximize his overall utility.

**Interaction Protocol.** Let $d \in \mathbb{N}$ denote the dimension of the problem and $\mathcal{A} \subseteq [-1, +1]^{d+1}$ the learner's action space[4]. Actions $\alpha \in \mathcal{A}$ correspond to hyperplanes, written in terms of their normal vectors, and we assume that the $(d+1)$-th coordinate encodes information about the intercept. Let $\mathcal{X} \subseteq ([0,1]^d, 1)$ the feature vector space, where by $([0,1]^d, 1)$ we denote the set of all $(d+1)$-dimensional vectors with values in $[0,1]$ in the first $d$ dimensions and value 1 in the $(d+1)$-th. Each

feature vector has an associated label $y \in \mathcal{Y} = \{-1, +1\}$. Formally, the interaction protocol (which repeats for $t \in [T]$) is given in Protocol 1, where by $\sigma_t$ we denote the tuple (feature vector, label).

---

**Protocol 1:** Learner-Agent Interaction at Round $t$

---

1   Nature adversarially selects feature vector $\mathbf{x}_t \in \mathcal{X} \subseteq ([0,1]^d, 1)$.      `// agent's original email`
2   The learner chooses action $\alpha_t \in \mathcal{A}$, and commits to it.      `// learner's linear classifier`
3   Agent observes $\alpha_t$ and $\sigma_t = (\mathbf{x}_t, y_t)$, where $y_t \in \mathcal{Y}$.      `// yt = +1, if non-spam originally`
4   Agent reports feature vector $\mathbf{r}_t(\alpha_t, \sigma_t) \in \mathcal{X}$ (potentially, $\mathbf{r}_t(\alpha_t, \sigma_t) \neq \mathbf{x}_t$).      `// altered email`
5   The learner observes $(\mathbf{r}_t(\alpha_t, \sigma_t), \hat{y}_t)$, where $\hat{y}_t \in \mathcal{Y}$ is the label of $\mathbf{r}_t(\alpha_t, \sigma_t)$, and incurs binary loss
    $\ell(\alpha_t, \mathbf{r}_t(\alpha_t, \sigma_t), \hat{y}_t) = \mathbb{1}\{\text{sgn}(\hat{y}_t \cdot \langle \alpha_t, \mathbf{r}_t(\alpha_t, \sigma_t) \rangle) = -1\}$.      `// loss on altered email`

---

**Agents' Behavior: $\delta$-Bounded Myopically Rational.** Drawing intuition from the email spam example, we focus on agents who can alter their feature vector *up to an extent* in order to make their email fool the classifier, and, if successful, they gain some value. The agent's reported feature vector $\mathbf{r}_t(\alpha_t, \sigma_t)$ is the solution to the following constrained optimization problem[5]:

$$\mathbf{r}_t(\alpha_t, \sigma_t) = \arg \max_{\|\mathbf{z} - \mathbf{x}_t\| \leq \delta} u_t(\alpha_t, \mathbf{z}, \sigma_t)$$

where $u_t(\cdot, \cdot, \cdot)$ is the agent's underlying utility function, *which is unknown to the learner*. In words, in choosing what to report, the agents are *myopic* (i.e., focus only on the current round $t$), *rational* (i.e., maximize their utility), and *bounded* (i.e., misreport in a ball of radius $\delta$ around $\mathbf{x}_t$). In such settings (e.g., email spam example), the agents derive no value if, by altering their feature vector from $\mathbf{x}_t$ to $\mathbf{r}_t(\alpha_t)$, they also change $y_t$. Indeed, a spammer ($y_t = -1$) wishes to fool the learner's classifier, without actually having to change their email to be a non-spam one ($\hat{y}_t = +1$). Since the agents are *rational* this means that the observed label by the learner is $\hat{y}_t = y_t$ and we only use notation $y_t$ for the rest of the paper. We call such agents $\delta$-Bounded Myopically Rational ($\delta$-BMR)[6].

Note that $\delta$-BMR agents include (but are not limited to!) a broad class of rational agents for strategic classification, like for example agents whose utility is defined as:

$$u_t(\alpha_t, \mathbf{r}_t(\alpha_t, \sigma_t), \sigma_t) = \delta' \cdot \mathbb{1}\{\text{sgn}(\langle \alpha_t, \mathbf{r}_t(\alpha_t, \sigma_t) \rangle) = +1\} - \|\mathbf{x}_t - \mathbf{r}_t(\alpha_t, \sigma_t)\|_2 \qquad (1)$$

where $\text{sgn}(x) = +1$ if $x \geq 0$ and $\text{sgn}(x) = -1$ otherwise. According to the utility presented in Eq. (1), the agents get a value of $\delta'$ if he gets labeled as $+1$ and incurs a cost (i.e., time/resources spent for altering the original $\mathbf{x}_t$) that is a metric. For this case, we have that $\delta \leq \delta'$.

**Model Comparison with Other Strategic Classification Works.** Learning in strategic classification settings was studied in an offline model by Hardt et al. [20], and subsequently, by Dong et al. [18] in an online model. Similar to our model, in [20, 18] the ground truth label $y_t$ remains *unchanged* even *after* the agent's manipulation. Moreover, the work of Dong et al. [18] is orthogonal to ours in one key aspect: they find the appropriate conditions which can guarantee that the best-response of an agent, written as a function of the learner's action, is concave. As a result, in their model the learner's loss function becomes *convex* and well known online convex optimization algorithms could be applied (e.g., [19, 12]) in conjunction with the mixture feedback that the learner receives. The foundation of our work, however, is settings with less smooth utility and loss functions for the agents and the learner respectively, where incompatibility issues arise. There has also been recent interest in strategic classification settings where the agents by misreporting actually end up changing their label $y_t$ [5, 32, 30]. These models are especially applicable in cases where in order to alter their feature vector $\mathbf{x}_t$ (e.g., qualifications for getting in college) the agents have to *improve* their ground truth label (e.g., actually *try* to become a better candidate [35]). In contrast, in our work we think of the misreports as "manipulations" that aim at gaming the system without altering $y_t$.

## 3   Stackelberg versus External Regret

For what follows, let $\{\alpha_t\}_{t=1}^T$ be the sequence of the learner's actions in a repeated Stackelberg game. The full proof of this section can be found in Appendix A.1, and Appendices A.2 and A.3 include detailed discussions around external and Stackelberg regret for learning in Stackelberg games.

We remark here that *incompatibility* results between *policy* and *external* regret were previously proven by Arora et al. [2], but the techniques used there are different and do not apply in our setting.

**Definition 3.1** (External). $R(T) = \sum_{t \in [T]} \ell(\alpha_t, \mathbf{r}_t(\alpha_t), y_t) - \min_{\alpha_E^\star \in \mathcal{A}} \sum_{t \in [T]} \ell(\alpha_E^\star, \mathbf{r}_t(\alpha_t), y_t)$.

The external regret compares the cumulative loss from $\{\alpha_t\}_{t \in [T]}$ to the cumulative loss incurred by the best-fixed action in hindsight, had learner *not* given the opportunity to the agents to best respond.

**Definition 3.2** (Stackelberg). $\mathcal{R}(T) = \sum_{t \in [T]} \ell(\alpha_t, \mathbf{r}_t(\alpha_t), y_t) - \min_{\alpha^\star \in \mathcal{A}} \sum_{t \in [T]} \ell(\alpha^\star, \mathbf{r}_t(\alpha^\star), y_t)$.

Stackelberg regret [4, 18] compares the loss from $\{\alpha_t\}_{t \in [T]}$ to the loss from the best-fixed action in hindsight, *had learner given the opportunity to the agents to best respond*.

**Theorem 3.3.** *There exists a repeated strategic classification setting against a $\delta$-BMR agent, where* every *action sequence from the learner with* sublinear *external regret incurs* linear *Stackelberg regret, and* every *action sequence for the learner with* sublinear *Stackelberg regret incurs* linear *external regret.*

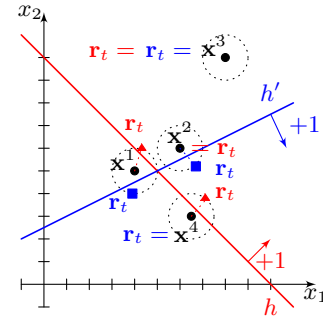

Figure 1: Black dots denote true feature vectors. Axes $x_1, x_2$ correspond to the two features. Dotted circles correspond to the $\delta$-bounded interval inside which agents can misreport. Blue squares correspond to misreports against action $h'$ and red triangles to misreports against action $h$.

*Proof Sketch.* We construct the following instance of an online strategic classification setting against $\delta$-BMR agents (pictorially shown in Figure 1). Let the action space be $\mathcal{A} = \{h, h'\}$ such that $h = (1, 1, -1)$ and $h' = (0.5, -1, 0.25)$, and let $\delta = 0.1$. Nature draws feature vectors $\mathbf{x}^1 = (0.4, 0.5, 1), \mathbf{x}^2 = (0.6, 0.6, 1), \mathbf{x}^3 = (0.8, 0.9, 1), \mathbf{x}^4 = (0.65, 0.3, 1)$ with probabilities $p^1 = 0.05, p^2 = 0.15, p^3 = 0.05, p^4 = 0.75$, and with labels $y^1 = -1, y^2 = -1, y^3 = +1, y^4 = +1$. Note that these original feature vectors are even *separable* by a *margin*! The expected loss for each action $\alpha \in \mathcal{A}$ corresponds to the number of mistakes that the learner makes against $\mathbf{r}_t(\alpha)$, which in turn depends on the probability with which nature drew each of the feature vectors $\mathbf{x}^1, \mathbf{x}^2, \mathbf{x}^3, \mathbf{x}^4$, e.g., $\mathbb{E}[\ell(h, \mathbf{r}_t(h), y_t)] = 0.2$ because classifier $h$ makes a mistake for points $\mathbf{x}^1$ and $\mathbf{x}^2$. Analogously, $\mathbb{E}[\ell(h', \mathbf{r}_t(h'), y_t)] = 0.25, \mathbb{E}[\ell(h, \mathbf{r}_t(h'), y_t)] = 0.9$ and $\mathbb{E}[\ell(h', \mathbf{r}_t(h), y_t)] = 0.05$.

Every action sequence that yields a sublinear Stackelberg regret in this instance, must include action $h$ *at least* $T - o(T)$ times (because $\mathbb{E}[\ell(h, \mathbf{r}_t(h), y_t)] < \mathbb{E}[\ell(h', \mathbf{r}_t(h'), y_t)]$), thus incurring cumulative *loss* $0.2(T - o(T)) + 0.9o(T)$. For such sequences, because responses $\mathbf{r}_t(h)$ appear at least $T - o(T)$ times, the best-fixed action in hindsight for the external regret is action $h'$, with cumulative loss: $0.05(T - o(T))$. This means that the external regret is at least $0.15T$, i.e., *linear*. For the next part of the proof, we show that if action $h'$ is played $T - o(T)$ times, then, the external regret is *sublinear*. This is enough to prove our theorem, since in this case the Stackelberg regret is $0.05T$, i.e., *linear*. If $h'$ is played $T - o(T)$ times, then the cumulative loss incurred is $0.05o(T) + 0.25(T - o(T))$ and the best fixed action in hindsight for the external regret is also action $h'$ with a cumulative loss $0.25(T - o(T)) + 0.05o(T)$. In other words, the external regret in this case is *sublinear*. ∎

## 4 The GRINDER Algorithm

In this section, we present GRINDER, an algorithm that learns to adaptively partition the learner's action space according to the agent's responses. To assist with the dense notation, we include notation tables in Appendix B.1. Formally, we prove the following[7] *data-dependent* performance guarantee.

**Theorem 4.1.** *Given a finite horizon $T$ the Stackelberg regret incurred by* GRINDER *(Algo. 2) is:*

$$\mathcal{R}(T) \leq \mathcal{O}\left(\sqrt{T \cdot \log\left(T \cdot \frac{\lambda(\mathcal{A})}{\lambda\left(\underline{p}_\delta\right)}\right) \cdot \log\left(\frac{\lambda(\mathcal{A})}{\lambda\left(\underline{p}_\delta\right)}\right)}\right)$$

*where by $\lambda(A)$ we denote the Lebesgue measure of any measurable space $A$, and by $\underline{p}_\delta$ we denote the polytope with the smallest Lebesgue measure that is induced by* GRINDER *after $T$ rounds' partition.*

**Inferring $\ell(\alpha, \mathbf{r}_t(\alpha), y_t)$ without Observing $\mathbf{x}_t$.** We think of the learner's and the agent's spaces as dual ones (Fig. 2), and focus on the agent's action space first. Since agents are $\delta$–BMR, then, for feature vector $\mathbf{x}_t$ the agent can only misreport within the ball of radius $\delta$ centered around $\mathbf{x}_t$, denoted by $\mathcal{B}_\delta(\mathbf{x}_t)$ (e.g., purple dotted circle in Fig. 2). Since the learner observes $\mathbf{r}_t(\alpha)$ and knows that the agent misreports in a ball of radius $\delta$ around $\mathbf{x}_t$ (which remains unknown to her), she knows that in the worst case the agent's $\mathbf{x}_t$ is found within the ball of radius $\delta$ centered at $\mathbf{r}_t(\alpha)$. This means that the set of all of the agent's possible misreports against any committed action $\alpha'$ from the learner $\mathbf{r}_t(\alpha')$ is the *augmented* $2\delta$ ball (e.g., green solid circle). Since $y_t$ is also observed by the learner, she can thus infer her loss $\ell(\alpha', \mathbf{r}_t(\alpha'), y_t)$ for *any* action $\alpha'$ that has $\mathcal{B}_{2\delta}(\mathbf{r}_t(\alpha))$ *fully* in one of its halfspaces (e.g., actions $\beta, \gamma$ in Fig. 2).

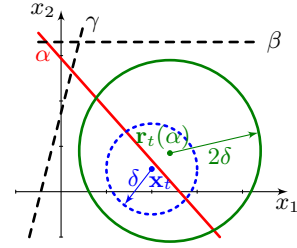

Figure 2: Agent's action space with axes $x_1, x_2$ corresponding to features. $\mathbf{x}_t$ is the agent's true feature vector and $\mathbf{r}_t(\alpha)$ his misreport against $\alpha$. Actions $\alpha, \beta, \gamma$ comprise the learner's action set.

In the learner's action space, actions $\alpha, \beta, \gamma$ are multidimensional *points*, and this has a nice mathematical translation. An action $\gamma$ has $\mathcal{B}_{2\delta}(\mathbf{r}_t(\alpha))$ fully in one of its halfspaces, if its distance from $\mathbf{r}_t(\alpha)$ is *more than* $2\delta$. Alternatively for actions $h \in \mathcal{A}$ such that:

$$\frac{|\langle h, \mathbf{r}_t(\alpha)\rangle|}{\|h\|_2} \leq 2\delta \Leftrightarrow |\langle h, \mathbf{r}_t(\alpha)\rangle| \leq 4\sqrt{d}\delta$$

where the last inequality comes from the fact that $\mathcal{A} \subseteq [-1, 1]^{d+1}$, the learner cannot infer $\ell(h, \mathbf{r}_t(h))$. But for all other actions $\gamma$ in $\mathcal{A}$, the learner can compute her loss $\ell(h, \mathbf{r}_t(h))$ *precisely*! From that, we derive that the learner can partition her action space into the following *polytopes*: upper polytopes $\mathcal{P}_t^u$, containing actions $\mathbf{w} \in \mathcal{A}$ such that $\langle \mathbf{w}, \mathbf{r}_t(\alpha)\rangle \geq 4\sqrt{d}\delta$ and lower polytopes $\mathcal{P}_t^l$, containing actions $\mathbf{w}' \in \mathcal{A}$ such that $\langle \mathbf{w}', \mathbf{r}_t(\alpha)\rangle \leq -4\sqrt{d}\delta$. The distinction into the two sets is helpful as one of them always assigns label $+1$ to the agent's best-response, and the other always assigns label $-1$. The sizes of $\mathcal{P}_t^u$ and $\mathcal{P}_t^l$ depend on $\delta$ and $\{\mathbf{x}_t\}_{t=1}^T$, but we omit these for the ease of notation.

**Algorithm Overview.** At each round $t$, GRINDER (Algo. 2) maintains a sequence of nested *polytopes* $\mathcal{P}_t$, with $\mathcal{P}_1 = \{\mathcal{A}\}$ and decides which action $\alpha_t$ to play according to a two-stage sampling process. We denote the resulting distribution by $\mathcal{D}_t$, and by $\mathbb{P}_t$ and $f_t(\alpha)$ the associated probability and probability density function.

After the learner observes $\mathbf{r}_t(\alpha_t)$, she computes two hyperplanes with the same normal vector $(\mathbf{r}_t(\alpha_t))$ and symmetric intercepts $(\pm 4\sqrt{d}\delta)$. These *boundary* hyperplanes are defined as:

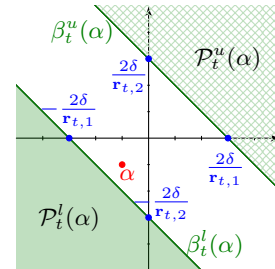

Figure 3: Polytope partitioning for $d = 2$. $\mathbf{r}_{t,1}, \mathbf{r}_{t,2}$ correspond to the $x_1$ and $x_2$ coordinates of $\mathbf{r}_t(\alpha)$.

$$\beta_t^u(\alpha_t) : \forall \mathbf{w} \in \mathcal{A}, \langle \mathbf{w}, \mathbf{r}_t(\alpha_t)\rangle = 4\sqrt{d}\delta$$
$$\beta_t^l(\alpha_t) : \forall \mathbf{w} \in \mathcal{A}, \langle \mathbf{w}, \mathbf{r}_t(\alpha_t)\rangle = -4\sqrt{d}\delta$$

and they split the learner's action space into three *regions*; one for which $\forall \mathbf{w} : \langle \mathbf{w}, \mathbf{r}_t(\alpha_t)\rangle \geq 4\sqrt{d}\delta$, one for which $\langle \mathbf{w}, \mathbf{r}_t(\alpha_t)\rangle \leq -4\sqrt{d}\delta$ and one for which $|\langle \mathbf{w}, \mathbf{r}_t(\alpha_t)\rangle| \leq 4\sqrt{d}\delta$ (see Fig. 3).

Let $H^+(\beta), H^-(\beta)$ denote the closed positive and negative halfspaces defined by hyperplane $\beta$ for intercept $4\sqrt{d}\delta$ and $-4\sqrt{d}\delta$ respectively[8]. Slightly abusing notation, we say that polytope $p \subseteq H^+(\beta)$ if for all actions $\alpha$ contained in $p$ it holds that $\alpha \in H^+(\beta)$.

We define action $\alpha_t$'s *upper* and *lower* polytopes sets to be the sets of polytopes such that $\mathcal{P}_t^u(\alpha_t) = \{p \subseteq \mathcal{P}_t, p \subseteq H^+(\beta_t^u(\alpha_t))\}$ and $\mathcal{P}_t^l(\alpha_t) = \{p \subseteq \mathcal{P}_t, p \subseteq H^-(\beta_t^l(\alpha_t))\}$ respectively. Defining these sets is useful since they represent the subsets of the learner's action space for which she can infer $\forall h : \ell(h, \mathbf{r}_t(h), y_t)$ *despite never observing* $\mathbf{x}_t$! To be more precise for $h \in \mathcal{P}_t^u(\alpha_t) \bigcup \mathcal{P}_t^l(\alpha_t)$:

$$\ell(h, \mathbf{r}_t(h), y_t) = \mathbb{1}\{y_t = -1\} \cdot \mathbb{1}\{h \in \mathcal{P}_t^u(\alpha_t)\} + \mathbb{1}\{y_t = +1\} \cdot \mathbb{1}\{h \in \mathcal{P}_t^l(\alpha_t)\}$$

**Algorithm 2:** GRINDER Algorithm for Strategic Classification

---
1   Initialize polytopes' set: $\mathcal{P}_0 = \{\mathcal{A}\}$.
2   Initialize polytope weights $w_1(p) = \lambda(p), p \in \mathcal{P}_0$.
3   Tune learning and exploration rates $\eta = \gamma \leq 1/2$, as specified in the analysis.
4   **for** $t \leftarrow 1$ **to** $T$ **do**
5     Compute $\forall p \in \mathcal{P}_t : \pi_t(p) = (1-\gamma)q_t(p) + \gamma \frac{\lambda(p)}{\lambda(\mathcal{A})}$.     `// distribution over polytopes`
     `/* Two-stage sampling: first, polytope, second, draw action from within.    */`
6     Select polytope $p_t \sim \pi_t$ from which you draw action $\alpha_t \sim \texttt{Unif}(p_t)$ and commit to $\alpha_t$.
7     Observe the agent's response $(\mathbf{r}_t(\alpha_t), y_t)$ to committed $\alpha_t$.
     `/* Space partitioning into smaller polytopes. `$\mathcal{P}_t$`: current polytopes set.    */`
8     Define a new set of polytopes $\mathcal{P}_{t+1} = \mathcal{P}_{t+1}^u(\alpha_t) \bigcup \mathcal{P}_{t+1}^m(\alpha_t) \bigcup \mathcal{P}_{t+1}^l(\alpha_t)$, where:
9     **for** *each polytope $p \in \mathcal{P}_t$* **do**
10       Add in $\mathcal{P}_{t+1}^u(\alpha_t)$ the non-empty intersection $p \bigcap H^+(\beta_t^u(\alpha_t))$   `// upper polytopes set`
11       Add in $\mathcal{P}_{t+1}^l(\alpha_t)$ the non-empty intersection $p \bigcap H^-(\beta_t^l(\alpha_t))$   `// lower polytopes set`
12       Add in $\mathcal{P}_{t+1}^m(\alpha_t)$ the non-empty remainder of $p$.     `// middle polytopes set`
13     Compute $\hat{\ell}(\alpha_t, \mathbf{r}_t(\alpha_t), y_t) = \frac{\ell(\alpha_t, \mathbf{r}_t(\alpha_t), y_t)}{\mathbb{P}_t^{\texttt{in}}[\alpha_t]}$.     `// loss estimator for chosen action`
14     **for** *each polytope $p \in \mathcal{P}_{t+1}$* **do**
      `/* upper and lower polytopes get full information                          */`
15       Compute $\hat{\ell}(p, \mathbf{r}_t(p), y_t) = \frac{\ell(p, \mathbf{r}_t(p), y_t) \cdot \mathbb{1}\{p \subseteq \mathcal{P}_{t+1}^u(\alpha_t) \bigcup \mathcal{P}_{t+1}^l(\alpha_t)\}}{\mathbb{P}_t^{\texttt{in}}[p]}$.
      `/* weight scaling with the Lebesgue measure of the polytope                 */`
16       Update $w_{t+1}(p) = \lambda(p) \exp\left(-\eta \sum_{\tau=1}^t \hat{\ell}(p, \mathbf{r}_t(p), y_t)\right), q_{t+1}(p) = \frac{w_{t+1}(p)}{\sum_{p' \in \mathcal{P}_{t+1}} w_{t+1}(p')}$.

---

The definition of the polytopes establishes that at each round the estimated loss within each polytope is *constant*. If a polytope has *not* been further "grinded" by the algorithm, then the estimated loss that was used to update the polytope has been the same within the actions of the polytope for each time step! This observation explains the way the weights of the polytopes are updated by scaling with the Lebesgue measure of each polytope. Due to the fact that the loss of all the points within a polytope is the same, we slightly abuse notation and we use $\ell(p, \mathbf{r}_t(p), y_t)$ to denote the loss for any action $\alpha \in p$ for round $t$, if the agent best-responded to it.

We next define the lower, upper, and middle $\sigma_t$-induced[9] polytope sets as: $\mathcal{P}_{t,\sigma_t}^l = \{\alpha \in p, p \in \mathcal{P}_t : \texttt{sdist}(\alpha, \mathbf{x}_t) \leq -2\delta\}$, $\mathcal{P}_{t,\sigma_t}^u = \{\alpha \in p, p \in \mathcal{P}_t : \texttt{sdist}(\alpha, \mathbf{x}_t) \geq 2\delta\}$, and $\mathcal{P}_{t,\sigma_t}^m = \{\alpha \in p, p \in \mathcal{P}_t : |\texttt{sdist}(\alpha, \mathbf{x}_t)| < 2\delta\}$, where $\texttt{sdist}(\alpha, \mathbf{x}_t) = \langle \alpha, \mathbf{x}_t \rangle / \|\alpha\|_2$.

GRINDER uses access to what we call an *in-oracle* (Def. 4.2). Our main regret theorem is stated for an accurate oracle, but we show that our regret guarantees still hold for approximation oracles (Lem. B.7). Such oracles can be constructed in practice, as we show in Sec. 4.1.

**Definition 4.2** (In-Oracle). *We define the* In-Oracle *as a black-box algorithm, which takes as input a polytope (resp. action) and returns the total* in-*probability for this polytope (resp. action):*

$$\mathbb{P}_t^{\texttt{in}}[p] = \int_{\mathcal{A}} \mathbb{P}_t\left[\left\{p \subseteq H^+\left(\beta_t^u(\alpha')\right)\right\} \bigcup \left\{p \subseteq H^-\left(\beta_t^l(\alpha')\right)\right\}\right] d\alpha'$$

We provide below the proof sketch for Thm 4.1. The full proof can be found in Appendix B. We also note that the algorithm can be turned into one that does not assume knowledge of $T$ or $\lambda\left(\underline{p}_\delta\right)$ by using the standard *doubling trick* [3].

*Proof Sketch of Thm 4.1.* Using properties of the pdf, we first prove that $\hat{\ell}(\cdot, \cdot, \cdot)$ is an unbiased estimator, and its variance is inversely dependent on quantity $\mathbb{P}_t^{\texttt{in}}[\alpha]$. Next, using the $\sigma_t$-induced polytopes sets, we can bound the variance of our estimator (Lem. B.4) by making a novel connection

with a graph theoretic lemma from the literature in online learning with feedback graphs ([1, Lem. 5], also stated in Lem. B.5):

$$
\mathop{\mathbb{E}}_{\alpha_t \sim \mathcal{D}_t} \left[ \frac{1}{\mathbb{P}_t^{\mathtt{in}}[\alpha_t]} \right] \leq 4 \log \left( \frac{4\lambda(\mathcal{A}) \cdot \left| \mathcal{P}_{t,\sigma_t}^u \cup \mathcal{P}_{t,\sigma_t}^l \right|}{\gamma \lambda \left( \underline{p}_\delta \right)} \right) + \lambda \left( \mathcal{P}_{t,\sigma_t}^m \right)
$$

To do so, we first expand the term $\mathbb{E}_{\alpha_t \sim \mathcal{D}_t} \left[ 1/\mathbb{P}_t^{\mathtt{in}}[\alpha_t] \right]$ as:

$$
\mathop{\mathbb{E}}_{\alpha_t \sim \mathcal{D}_t} \left[ \frac{1}{\mathbb{P}_t^{\mathtt{in}}[\alpha_t]} \right] = \int_{\mathcal{A}} \frac{f_t(\alpha)}{\mathbb{P}_t^{\mathtt{in}}[\alpha]} d\alpha = \underbrace{\int_{\bigcup \left( \mathcal{P}_{t,\sigma_t}^u \cup \mathcal{P}_{t,\sigma_t}^l \right)} \frac{f_t(\alpha)}{\mathbb{P}_t^{\mathtt{in}}[\alpha]} d\alpha}_{Q_1} + \underbrace{\int_{\bigcup \mathcal{P}_{t,\sigma_t}^m} \frac{f_t(\alpha)}{\mathbb{P}_t^{\mathtt{in}}[\alpha]} d\alpha}_{Q_2} \tag{2}
$$

Due to the fact that GRINDER uses *conservative* estimates of the *true* action space with $\mathtt{sdist}(\alpha, \mathbf{x}_t) \leq \delta$, term $Q_2$ can be upper bounded by $\lambda(\mathcal{P}_{t,\sigma_t}^m)$. Upper bounding $Q_1$ is significantly more involved (Lem. B.4). First, observe that each of the actions in $\mathcal{P}_{t,\sigma_t}^u, \mathcal{P}_{t,\sigma_t}^l$ gets updated with probability 1 by any other action in the sets $\mathcal{P}_{t,\sigma_t}^u, \mathcal{P}_{t,\sigma_t}^l$. This is because for any of the actions in $\mathcal{P}_{t,\sigma_t}^u, \mathcal{P}_{t,\sigma_t}^l$, the agent could not have possibly misreported. So, for all actions $\alpha \in \mathcal{P}_{t,\sigma_t}^u \cup \mathcal{P}_{t,\sigma_t}^l$ we have that: $\mathbb{P}_t^{\mathtt{in}}[\alpha] \geq \sum_{p \in \mathcal{P}_{t,\sigma_t}^u \cup \mathcal{P}_{t,\sigma_t}^l} \pi_t(p)$. As a result, we can instead think about the set of *polytopes* that belong in $\mathcal{P}_{t,\sigma_t}^u$ and $\mathcal{P}_{t,\sigma_t}^l$ as forming a fully connected feedback graph. The latter, coupled with the fact that our exploration term makes sure that each polytope $p$ is chosen with probability at least $\lambda(p)/\lambda(\mathcal{A})$ gives that: $Q_1 \leq 4 \log \left( \frac{4\lambda(\mathcal{A}) \cdot \left| \mathcal{P}_{t,\sigma_t}^u \cup \mathcal{P}_{t,\sigma_t}^l \right|}{\lambda(\underline{p}_\delta) \cdot \gamma} \right)$.

Expressing everything in terms of polytopes rather than individual actions is *critical* in the previous step, because applying [1, Lem. 5] on $\mathcal{A}$, rather than $\mathcal{P}_t$, gives vacuous regret upper bounds, due to the logarithmic dependence in the number of nodes of the feedback graph, which is infinite for the case of $\mathcal{A}$. The penultimate step of the proof (Lem. B.6) is a second order regret bound for GRINDER on the estimated losses $\hat{\ell}(\cdot, \cdot)$, which should be viewed as the continuous variant of the standard discrete-action second order regret bound for EXP3-type algorithms. In order to derive the bound stated in Thm 4.1 we upper bound the total number of $\sigma_t$-induced polytopes with $\lambda(\mathcal{A})/\lambda(\underline{p}_\delta)$. ∎

The regret guarantee of GRINDER is *preserved* if instead of an *accurate* in-oracle it is provided an *ε-approximate* one, where $\varepsilon \leq 1/\sqrt{T}$ (Lemma B.7). As we also validate in Sec. 4.1, in settings where few points violate the margin between the $+1$ and the $-1$ labeled points such approximation oracles do exist and are relatively easy to construct.

Computing the volume of polytopes is a #-P hard problem, so GRINDER should be viewed as an information-theoretic result. However, if GRINDER is provided access to an efficient black-box algorithm for computing the volume of a polytope, its runtime complexity is $\mathcal{O}(T^d)$ (Lem. B.8).

## 4.1 Simulations

In this subsection we present our simulation results[10]. We build simulation datasets since in order to *evaluate* the performance of our algorithms one needs to know the original datapoints $\mathbf{x}_t$. The results of our simulations are presented in Fig. 4.

For the simulation, we run GRINDER against EXP3 for a horizon $T = 1000$, where each round was repeated for 30 repetitions. The $\delta$-BMR agents that we used are best-responding according to the utility function of Eq. (1), and we studied 5 different values for $\delta$: $0.05, 0.1, 0.15, 0.3, 0.5$. The $+1$ labeled points are drawn from Gaussian distribution as $\mathbf{x}_t \sim (\mathcal{N}(0.7, 0.3), \mathcal{N}(0.7, 0.3))$ and the $-1$ labeled points are drawn from $\mathbf{x}_t \sim (\mathcal{N}(0.4, 0.3), \mathcal{N}(0.4, 0.3))$. Thus we establish that for the majority of the points there is a clear "margin" but there are few points that violate it (i.e., there exists no perfect linear classifier).

EXP3 is always run with a fixed set of actions and always suffers a dependence on the different actions (i.e., not $\delta$). We then run GRINDER in the same fixed set of actions and with a continuous

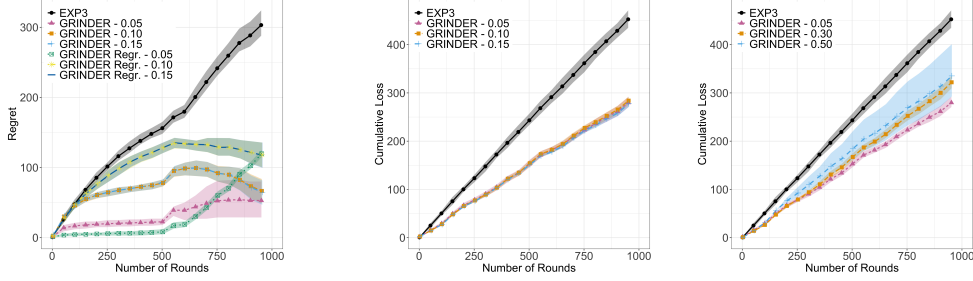

Figure 4: Performance of GRINDER vs. EXP3. In all cases, GRINDER outperforms EXP3. Solid lines correspond to average regret/loss, and opaque bands correspond to 10th and 90th percentile. **Left:** discrete action sets, accurate and regression oracle. **Middle:** Continuous action set for GRINDER with $\delta = 0.05, 0.10, 0.15$. **Right:** Continuous action set for GRINDER with $\delta = 0.05, 0.3, 0.5$.

action set. For the discrete action set, we include the results for both the accurate and the regression-based approximate oracle. We remark that if the action set is discrete, then GRINDER becomes similar to standard online learning with feedback graph algorithms (see e.g., [1]), but the feedback graph is built according to $\delta$-BMR agents. In this case, the regret scales as $\mathcal{O}(a(G)\log T)$, where $a(G)$ is the independence number of graph $G$.

For the continuous action set it is not possible to identify the best-fixed action in hindsight. As a result, we report the cumulative loss. In Appendix D, we include additional simulations for a different family of $\delta$-BMR agents[11], and different distributions of labels.

In order to build the approximation oracle we used past data and we trained a logistic regression model for each polytope, learning the probability that it is updated. Our model has "recency bias" and gives more weight to more recent datapoints. We expect that for more accurate oracles, our results are strengthened, as proved by our theoretical bounds.

Validating our theoretical results, GRINDER outperforms the benchmark, *despite the fact that we use an approximation oracle*. We also see that in the discrete action set, where an accurate oracle can be constructed, GRINDER performs much better than the regression oracle. As expected, GRINDER's performance becomes worse as the power of the adversary increases (i.e., as $\delta$ grows larger).

**Why not compare GRINDER with a convex surrogate?** In this paragraph, we explain our decision to only compare GRINDER with EXP3. In fact, *no standard convex surrogate can be used* for learning linear classifiers against $\delta$-BMR, since the learner does not know precisely the agent's response *function* $\mathbf{r}_t(\alpha)$. As a result, the learner cannot guarantee that $\ell(\alpha, \mathbf{r}_t(\alpha))$ is *convex* in $\alpha$, even if $\ell(\alpha, \mathbf{z})$ ($\mathbf{z}$ being independent of $\alpha$) is convex in $\alpha$! Concretely, think about the following counterexample: let $h = (1, 1, -1), h' = (0.5, -1, 0.25)$ be two hyperplanes, a point $\mathbf{x} = (0.55, 0.4), y = +1$, $\delta = 0.1$, and let $\ell(h, \mathbf{r}(h)) = \max\{0, 1 - y \cdot \langle h, \mathbf{r}(h) \rangle\}$ (i.e., hinge loss, which is convex). We show that when $(\mathbf{x}, y)$ is a $\delta$-BMR agent, $\ell(\alpha, \mathbf{r}(\alpha))$ is *no longer convex* in $\alpha$. Take $b = 0.5$ and construct $h_b = 0.5h + 0.5h' = (0.75, 0, -0.375) = (1, 0, -0.5)$. $(\mathbf{x}, y)$ only misreports to (say) $(0.61, 0.4)$ when presented with $h$ (as $h_b$ and $h'$ classify $\mathbf{x}$ as $+1$). Computing the

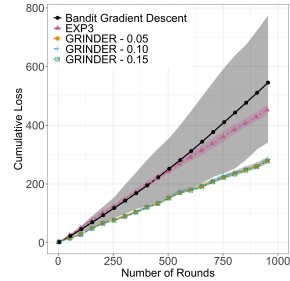

Figure 5: Performance of GRINDER vs. BGD.

loss: $\ell(h_b, \mathbf{r}(h_b)) = 0.95, \ell(h, \mathbf{r}(h)) = 0.99$ and $\ell(h', \mathbf{r}(h')) = 0.875$, so, $\ell(h_b, \mathbf{r}(h_b)) > b\ell(h, \mathbf{r}(h)) + (1 - b)\ell(h', \mathbf{r}(h'))$. Since in general $\ell(\alpha, \mathbf{r}(\alpha))$ is not convex, it is may seem unfair to compare Bandit Gradient Descent (BGD) with GRINDER but we include comparison results in Fig. 5, where GRINDER greatly outperforms BGD, for completeness. Identifying surrogate losses that are convex against $\delta$-BMR agents remains a very interesting open question.

# 5  Lower Bound

In this section we prove nearly matching lower bounds for learning a linear classifier against $\delta$–BMR agents. To do so, we use the geometry of the sequence of datapoints $\sigma_t$ interpreted in the dual space. The proofs can be found in Appendix C.

**Theorem 5.1.** *For any strategy and any $\delta$, there exists a sequence of $\{\sigma_t\}_{t=1}^{T}$ such that:*

$$\mathbb{E}\left[\sum_{t\in[T]}\ell(\alpha_t,\mathbf{r}_t(\alpha_t),y_t)\right] - \min_{\alpha^\star\in\mathcal{A}}\mathbb{E}\left[\sum_{t\in[T]}\ell(\alpha^\star,\mathbf{r}_t(\alpha^\star),y_t)\right] \geq \frac{1}{9\sqrt{2}}\sqrt{T\log\left(\frac{\lambda(\mathcal{A})}{\lambda(\tilde{p}_\delta)}\right)} \quad (3)$$

*where $\tilde{p}_\delta$ is the* smallest $\sigma_t$-induced polytope from the sequence of $\{\sigma_t\}_{t=1}^{T}$.

We remark here that that the $\sigma_t$-induced polytopes as defined in the previous section depend *only* on the sequence of agents that we face, and not on the properties of any algorithm. We proceed with the proof sketch of our lower bound.

*Proof Sketch.* Fix a $\delta > 0$, and assume that the agents are *truthful*[12] (i.e., $\mathbf{r}_t(\alpha) = \mathbf{x}_t, \forall t \in [T], \forall \alpha \in \mathcal{A}$). Faithful to our model, however, the learner can only observe $\mathbf{r}_t(\alpha)$, without knowing its equivalence to $\mathbf{x}_t$. We prove the theorem in two steps.

In the first step (Lem. C.1) we show a more relaxed lower bound of order $\Omega(\sqrt{T})$. To prove this, we fix a particular feature vector $\mathbf{x}$ for the agent, and two different adversarial environments (call them $U$ and $L$) choosing the label of $\mathbf{x}$ according to different Bernoulli probability distributions; one of them favors label $y_t = +1$, while the other favors label $y_t = -1$. The $\Omega(\sqrt{T})$ lower bound corresponds to the regret accrued by the learner in order to distinguish between $U$ and $L$.

For the second step, we separate the horizon into $\Phi = \log(\lambda(\mathcal{A})/\lambda(\tilde{p}_\delta))$ phases, each comprised by $T/\Phi$ consecutive rounds. In all rounds of a phase, the agent has the same $\mathbf{x}$ and the labels are constructed by adversarial environments $U$ and $L$. At the end of each phase, either $U$ or $L$ must have caused regret at least $\Omega(\sqrt{T/\Phi})$. According to which one it was, nature selects the feature vector for the next phase in a way that guarantees that *one* of the best-fixed actions for all previous phases is *still part of the optimal actions* at this phase. The general pattern that we follow for the feature vectors of each phase is $\mathbf{x}_\phi = \left(\frac{1}{2}, \frac{1}{4}\left(1 + \kappa_\phi \cdot 2^\phi\right)\right)$ where $\kappa_\phi$ is a phase-specific constant for which $\kappa_0 = 1$ and $\kappa_{\phi+1} = 2\kappa_\phi + 1$ if the environment causing regret $\Omega(\sqrt{T})$ was $U$ and $\kappa_{\phi+1} = 2\kappa_\phi - 1$ otherwise. This pattern establishes that the feature vectors are spaced in a way that every algorithm would be penalized enough, in order to be able to discern their labels. ∎

## Acknowledgments and Funding Disclosures

The authors are grateful to Kira Goldner, Akshay Krishnamurthy, Thodoris Lykouris, David Parkes, and Vasilis Syrgkanis for helpful discussions at different stages of this work, and to the anonymous reviewers for their comments and suggestions.

This work was partially supported by the National Science Foundation under grants CCF-1718549 and IIS-2007951 and the Harvard Data Science Initiative.

## Broader Impact

In this paper we put forth and study a model for strategic classification against real-life agents. We believe that when Machine Learning (ML) algorithms are deployed for real-life decision-making, the agents that we face (e.g., the human subjects that we wish to "classify") will not try to sabotage the decision-making algorithms, but they will try to manipulate them. The power and the ability to manipulate a decision-making rule is inherently tied to the learned outcome for the rule. Strategic ML aims to address these concerns with the construction of ML algorithms that are either *strategyproof* (i.e., they provide no incentive to the agents to manipulate) or that are *strategy-aware* (i.e., they are not affected too much by strategic manipulations).

We think that since classification is a fundamental ML task, computer scientists together with economists should try to understand two key questions:

1. What is the "correct" behavioral model to explain the agents' reports?
2. How do we design ML algorithms that take this behavioral model into account, before deploying the algorithms for policy making?

However, blindly optimizing predictive accuracy and the role of incentives in it, can have detrimental societal effects. For example, there have been works recently analyzing the disparate effects that strategic manipulation can have among different groups ([21, 29]) when the agents' utility functions are similar to the ones considered by Hardt et al. [20]. We believe that a next step for learning strategy-aware linear classifiers against $\delta$–BMR is to study whether there exist algorithms that concurrently satisfy the no-regret property and provide better societal guarantees.

## Footnotes

[1]We refer to the learner as a female (she/her/hers) and to the agents as male (he/his/him).

[2]Due to the logarithmic dependence on the number of actions.

[3]Even though the formal definition of Stackelberg regret was only later introduced by Dong et al. [18].

[4]This is wlog, as the normal vector of any hyperplane can be normalized to lie in $[-1, 1]^{d+1}$.

[5] For simplicity, we denote $\mathbf{r}_t(\alpha_t) = \mathbf{r}_t(\alpha_t, \sigma_t)$ when clear from context.

[6] Note that if the agents were *adversarial*, they would report $\mathbf{r}(\alpha_t) = \arg \max_{\|\mathbf{z} - \mathbf{x}_t\| \leq \delta} \ell(\alpha, \mathbf{z}, y_t)$.

[7] Our actual bound is *tigher*, but harder to interpret without analyzing the algorithm.

[8] i.e., $\alpha \in H^+(\beta)$ if $\alpha \in \mathcal{A}, \langle \beta, \alpha\rangle \geq 4\sqrt{d}\delta$ and similarly, $\alpha \in H^-(\beta)$ if $\alpha \in \mathcal{A}, \langle \beta, \alpha\rangle \leq -4\sqrt{d}\delta$

[9]These can only be computed if one has access to the agent's true datapoint $\sigma_t = (\mathbf{x}_t, y_t)$. However, we only use them in our analysis, and GRINDER does not require access to them.

[10]Our code is publicly available here: https://github.com/charapod/learn-strat-class

[11]Namely, their utility function is: $u_t(\alpha_t, \mathbf{r}_t(\alpha_t), \sigma_t) = \delta' \cdot \langle \alpha_t, \mathbf{r}_t(\alpha_t) \rangle - \|\mathbf{x}_t - \mathbf{r}_t(\alpha_t)\|_2$.

[12]Truthful agents are $\delta$–BMR agents, so the lower bound holds for the whole family of $\delta$–BMR agents.

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
