[Supplementary Material]

# Supplementary Material for Paper ID: 4406

## A   Appendix for Section 3

### A.1   Missing Proofs from Section 3

*Proof of Theorem 3.3.* For completeness, we outline again the instance described earlier. Let an action space $\mathcal{A} = \{h, h'\}$ such that $h = (1, 1, -1)$ and $h' = (0.5, -1, 0.25)$, and let $\delta = 0.1$. The environment draws feature vectors $\mathbf{x}^1 = (0.4, 0.5), \mathbf{x}^2 = (0.6, 0.6), \mathbf{x}^3 = (0.8, 0.9), \mathbf{x}^4 = (0.65, 0.3)$ with probabilities $p^1 = 0.05, p^2 = 0.15, p^3 = 0.05, p^4 = 0.75$ respectively, and with labels $y^1 = -1, y^2 = -1, y^3 = +1, y^4 = +1$. For clarity, Figure 1 provides a pictorial depiction of the example, along with the best responses of the agents for each action. We first explain the values that the loss function takes according to the best-responses of the agents and the feature vectors drawn by nature.

- $(\mathbb{E}\left[\ell\left(h, \mathbf{r}_t\left(h\right), y_t\right)\right])$ When the learner plays $h$ against agent's responses $\mathbf{r}_t\left(h\right)$, she makes a mistake in her prediction every time that the environment drew $\mathbf{x}_1$ or $\mathbf{x}_2$ for the agent. This is because for $\mathbf{x}^1$ the agent can misreport and fool the hyperplane. For $\mathbf{x}^2$ the agent does not need to misreport; hyperplane $h$ classifies it as $+1$ erroneously already. For $\mathbf{x}^4$ the agent can misreport and get correctly classified and for $\mathbf{x}^3$ the hyperplane is correct all by itself. Hence:

$$\mathbb{E}\left[\ell\left(h, \mathbf{r}_t\left(h\right), y_t\right)\right] = \mathbb{P}\left[\text{nature draws } \mathbf{x}^1 \text{ or } \mathbf{x}^2\right] = p^1 + p^2 = 0.2$$

- $(\mathbb{E}\left[\ell\left(h', \mathbf{r}_t\left(h'\right), y_t\right)\right])$ When the learner plays $h'$ against agent's responses $\mathbf{r}_t\left(h'\right)$, she makes a mistake in her prediction every time that the environment drew $\mathbf{x}_1$ or $\mathbf{x}_2$ or $\mathbf{x}_3$ for the agent. This is because for both $\mathbf{x}^1$ and $\mathbf{x}^2$ the agent could misreport and fool the hyperplane and for $\mathbf{x}^3$ the hyperplane classifies it incorrectly, but there is nothing that the learner can do to change it (due to $\delta$-boundedness). For $\mathbf{x}^4$ the hyperplane classifies the point correctly, without the need of misreport from the agent. Hence:

$$\mathbb{E}\left[\ell\left(h', \mathbf{r}_t\left(h'\right), y_t\right)\right] = \mathbb{P}\left[\text{nature draws } \mathbf{x}^1 \text{ or } \mathbf{x}^2 \text{ or } \mathbf{x}^3\right] = p^1 + p^2 + p^3 = 0.25$$

- $(\mathbb{E}\left[\ell\left(h, \mathbf{r}_t\left(h'\right), y_t\right)\right])$ When the learner plays $h$ against agent's responses $\mathbf{r}_t\left(h'\right)$, she makes a mistake in her prediction every time that the environment drew $\mathbf{x}_2$ or $\mathbf{x}_4$ for the agent, i.e.,

$$\mathbb{E}\left[\ell\left(h, \mathbf{r}_t\left(h'\right), y_t\right)\right] = \mathbb{P}\left[\text{nature draws } \mathbf{x}^2 \text{ or } \mathbf{x}^4\right] = p^2 + p^4 = 0.9$$

- $(\mathbb{E}\left[\ell\left(h', \mathbf{r}_t\left(h\right), y_t\right)\right])$ When the learner plays $h'$ against agent's responses $\mathbf{r}_t\left(h\right)$, she makes a mistake in her prediction every time that the environment drew $\mathbf{x}_3$ for the agent, i.e.,

$$\mathbb{E}\left[\ell\left(h', \mathbf{r}_t\left(h\right), y_t\right)\right] = \mathbb{P}\left[\text{nature draws } \mathbf{x}^3\right] = p^3 = 0.05$$

We now prove that *any* sequence with sublinear Stackelberg regret will have linear external regret. Observe that for the Stackelberg regret, the best fixed action in hindsight is action $h$, with cumulative loss $0.2T$. Therefore, any action sequence that yields sublinear Stackelberg regret must have cumulative loss $0.2T + o(T)$, meaning that action $h'$ is played at most $o(T)$ times, while action $h$ is played at least $T - o(T)$ times. Given this, we proceed by identifying the best fixed action for the *external* regret in any action such sequence $\{\alpha_t\}_{t=1}^{T}$. For that, we compute the loss that any of the actions in $\mathcal{A}$ would incur, had they been the fixed action for sequence $\{\alpha_t\}_{t=1}^{T}$.

Assume that action $h$ was the fixed action in hindsight for the sequence $\{\alpha_t\}_{t=1}^{T}$. Then, the cumulative loss incurred by playing $h$ constantly for $T$ rounds, denoted by $\sum_{t=1}^{T} \ell\left(h, \mathbf{r}_t(\alpha_t), y_t\right)$ is:

$$\underbrace{0.2(T - o(T))}_{\substack{\text{loss incurred when playing} \\ h \text{ against } \mathbf{r}_t(h)}} + \underbrace{0.9 o(T)}_{\substack{\text{loss incurred when playing} \\ h \text{ against } \mathbf{r}_t(h')}}$$

Assume that action $h'$ was the fixed action in hindsight for the aforementioned action sequence. Then, the cumulative loss incurred by playing $h'$, denoted by $\sum_{t=1}^{T} \ell(h', \mathbf{r}_t(\alpha_t), y_t)$ is equal to

$$\underbrace{0.05(T - o(T))}_{\substack{\text{loss incurred when playing} \\ h' \text{ against } \mathbf{r}_t(h)}} + \underbrace{0.25 o(T)}_{\substack{\text{loss incurred when playing} \\ h' \text{ against } \mathbf{r}_t(h')}}$$

Hence, we have that the best fixed action in hindsight for the external regret for the sequence $\{\alpha_t\}_{t=1}^{T}$ is action $h'$. This means, however, that for the sequence $\{\alpha_t\}_{t=1}^{T}$, which guaranteed sublinear Stackelberg regret, the external regret is *linear* in $T$:

$$R(T) \geq 0.2T - 0.05T \geq 0.15T$$

Moving forward, we prove that *any* action sequence with sublinear external regret will have linear Stackelberg regret. Since we previously proved that any action sequence $\{\alpha_t\}_{t=1}^{T}$ with sublinear Stackelberg regret plays at least $T - o(T)$ times action $h$ and this resulted in having linear external regret, we only need to consider sequences where action $h'$ is played $T - o(T)$ times, while action $h$ is played for $o(T)$ times. For any such action sequence, it suffices to show that the external regret will be sublinear, since for any such sequence the Stackelberg regret will be linear:

$$\mathcal{R}(T) = 0.2 o(T) + 0.25 \cdot (T - o(T)) - 0.2T \geq 0.05T$$

Similarly to the analysis above, we distinguish the following cases. Assume that action $h$ was the fixed action in hindsight for $\{\alpha_t\}_{t=1}^{T}$. Then, the cumulative loss incurred by playing $h$ is $\sum_{t=1}^{T} \ell(h, \mathbf{r}_t(\alpha_t), y_t) = 0.2 o(T) + 0.9(T - o(T))$. Assume that action $h'$ was the fixed action in hindsight for the aforementioned action sequence. Then, the cumulative loss incurred by playing $h'$ is $\sum_{t=1}^{T} \ell(h', \mathbf{r}_t(\alpha_t), y_t) = 0.05 o(T) + 0.25(T - o(T))$. As a result, the best fixed action in hindsight for the Stackelberg regret would be action $h'$, yielding external regret $o(T)$, i.e., sublinear. This concludes our proof. ∎

## A.2 Purely Adversarial and Cooperative Stackelberg Games

Despite the worst-case incompatibility results that we have shown for the notions of external and Stackelberg regret, there are families of repeated games for which there is a clear *hierarchy* between the two. In this subsection, we study two of the most important ones; the family of *Purely Adversarial*, and the family of *Purely Cooperative Stackelberg Games*.

**Definition A.1** (Purely Adversarial Stackelberg Game (PASGs)). *We call a Stackelberg Game Purely Adversarial, if for all actions $\alpha' \in \mathcal{A}$ for the loss of the learner it holds that: $\ell(\alpha, \mathbf{r}(\alpha), y_t) \geq \ell(\alpha, \mathbf{r}(\alpha'), y_t)$, i.e., the agent inflicts the* highest *loss to the learner, when best-responding to the action to which she committed.*

**Definition A.2** (Purely Cooperative Stackelberg Game (PCSGs)). *We call a Stackelberg Game Purely Cooperative if for all actions $\alpha' \in \mathcal{A}$ for the loss of the learner it holds that: $\ell(\alpha, \mathbf{r}(\alpha), y_t) \leq \ell(\alpha, \mathbf{r}(\alpha'), y_t)$, i.e., the agent inflicts the* lowest *loss to the learner, when best-responding to the action to which she committed.*

We remark here that despite their similarities, PASGs and PCSGs are *not* equivalent to zero-sum games; in fact, it is easy to see that every zero-sum game is either a PASG or a PCSG, but the converse is *not* true (see e.g., the example loss matrix given in Table 1 where the first coordinate of tuple $(i, j)$ corresponds to the loss of the learner, and the second to the loss of the agent). Next, we outline the hierarchy between external and Stackelberg regret in repeated PASGs and PCSGs.

|  | $\mathbf{r}_t(\alpha)$ | $\mathbf{r}_t(\alpha')$ |
|---|---|---|
| $\alpha$ | $(7, -1)$ | $(6, -3)$ |
| $\alpha'$ | $(6, -3)$ | $(7, -1)$ |

Table 1: Example of a PASG that is not zero-sum.

**Lemma A.3.** *In repeated PASGs, Stackelberg regret is upper bounded by external regret, i.e., $\mathcal{R}(T) \leq R(T)$. In other words, any no-Stackelberg regret sequence of actions is also a no-external regret one.*

*Proof.* Let $\tilde{\alpha} = \arg\min_{\alpha \in \mathcal{A}} \sum_{t=1}^{T} \ell(\alpha, \mathbf{r}_t(\alpha_t), y_t)$ and $\alpha^\star = \arg\min_{\alpha \in \mathcal{A}} \sum_{t=1}^{T} \ell(\alpha, \mathbf{r}_t(\alpha), y_t)$. Then:

$$
\begin{aligned}
R(T) &= \sum_{t=1}^{T} \ell(\alpha_t, \mathbf{r}_t(\alpha_t), y_t) - \sum_{t=1}^{T} \ell(\tilde{\alpha}, \mathbf{r}_t(\alpha_t), y_t) && \text{(definition of external regret)} \\
&\geq \sum_{t=1}^{T} \ell(\alpha_t, \mathbf{r}_t(\alpha_t), y_t) - \sum_{t=1}^{T} \ell(\alpha^\star, \mathbf{r}_t(\alpha_t), y_t) && \text{(definition of } \tilde{\alpha}) \\
&\geq \sum_{t=1}^{T} \ell(\alpha_t, \mathbf{r}_t(\alpha_t), y_t) - \sum_{t=1}^{T} \ell(\alpha^\star, \mathbf{r}_t(\alpha^\star), y_t) && (\ell(\alpha^\star, \mathbf{r}_t(\alpha_t), y_t) \leq \ell(\alpha^\star, \mathbf{r}_t(\alpha^\star), y_t)) \\
&= \mathcal{R}(T)
\end{aligned}
$$

∎

On the other hand, for PCSGs it holds that:

**Lemma A.4.** *In repeated PCSGs, Stackelberg regret is lower bounded by external regret, i.e., $\mathcal{R}(T) \geq R(T)$. In other words, any no-external regret sequence of actions is also a no-Stackelberg regret one.*

*Proof.* Let $\tilde{\alpha} = \arg\min_{\alpha \in \mathcal{A}} \sum_{t=1}^{T} \ell(\alpha, \mathbf{r}_t(\alpha_t), y_t)$ and $\alpha^\star = \arg\min_{\alpha \in \mathcal{A}} \sum_{t=1}^{T} \ell(\alpha, \mathbf{r}_t(\alpha), y_t)$. Then:

$$
\begin{aligned}
R(T) &= \sum_{t=1}^{T} \ell(\alpha_t, \mathbf{r}_t(\alpha_t), y_t) - \sum_{t=1}^{T} \ell(\tilde{\alpha}, \mathbf{r}_t(\alpha_t), y_t) && \text{(definition of external regret)} \\
&\leq \sum_{t=1}^{T} \ell(\alpha_t, \mathbf{r}_t(\alpha_t), y_t) - \sum_{t=1}^{T} \ell(\tilde{\alpha}, \mathbf{r}_t(\tilde{\alpha}), y_t) && \text{(definition of PCSGs)} \\
&\leq \sum_{t=1}^{T} \ell(\alpha_t, \mathbf{r}_t(\alpha_t), y_t) - \sum_{t=1}^{T} \ell(\alpha^\star, \mathbf{r}_t(\alpha^\star), y_t) && \text{(definition of } \alpha^\star) \\
&= \mathcal{R}(T)
\end{aligned}
$$

∎

## A.3 The Function $\ell(\alpha, \mathbf{r}_t(\alpha), y_t)$

As we mentioned in the main body, the learner's loss function $\ell(\alpha, \mathbf{r}_t(\alpha), y_t)$ is generally not Lipschitz in her chosen action $\alpha$. For that, we study below the quantity $|\ell(\alpha, \mathbf{r}_t(\alpha), y_t) - \ell(\alpha', \mathbf{r}_t(\alpha'), y_t)|$.

**Lemma A.5.** *Let $\ell(x, y, z)$ denote the learner's loss function in a Stackelberg game, such that $\ell$ is $L_1$-Lipschitz with respect to the first argument, and $L_2$-Lipschitz with respect to the second. Then, for the learner's loss between any two actions $\alpha, \alpha' \in \mathcal{A}$ it holds that:*

$$|\ell(\alpha, \mathbf{r}_t(\alpha), y_t) - \ell(\alpha', \mathbf{r}_t(\alpha'), y_t)| \leq \max\left\{L_1 \cdot \|\alpha' - \alpha\|, L_2 \cdot \|\mathbf{r}_t(\alpha) - \mathbf{r}_t(\alpha')\|\right\}$$

*Proof.* We split the set of actions $\mathcal{A}$ into pairs $(\alpha, \alpha')$ satisfying the following properties:

1. For pair $(\alpha, \alpha')$ it holds that: $\ell(\alpha, \mathbf{r}_t(\alpha), y_t) \geq \ell(\alpha, \mathbf{r}_t(\alpha'), y_t)$ and $\ell(\alpha', \mathbf{r}_t(\alpha'), y_t) \geq \ell(\alpha', \mathbf{r}_t(\alpha), y_t)$. In other words, by best-responding the agent causes the biggest loss to the learner. Observe that, given that $\ell$ is $L_1$-Lipschitz in its first argument, we have that:

   $$\ell(\alpha', \mathbf{r}_t(\alpha'), y_t) - \ell(\alpha, \mathbf{r}_t(\alpha), y_t) \geq \ell(\alpha', \mathbf{r}_t(\alpha), y_t) - \ell(\alpha, \mathbf{r}_t(\alpha), y_t) \geq -L_1\|\alpha' - \alpha\|$$

   and

   $$\ell(\alpha', \mathbf{r}_t(\alpha'), y_t) - \ell(\alpha, \mathbf{r}_t(\alpha), y_t) \leq \ell(\alpha', \mathbf{r}_t(\alpha'), y_t) - \ell(\alpha, \mathbf{r}_t(\alpha'), y_t) \leq L_1\|\alpha' - \alpha\|$$

   Therefore, for such pairs of actions function $\ell(\alpha, \mathbf{r}_t(\alpha), y_t)$ is $L_1$-Lipschitz with respect to $\alpha$.

2. For pair $(\alpha, \alpha')$ it holds that: $\ell(\alpha, \mathbf{r}_t(\alpha), y_t) \leq \ell(\alpha, \mathbf{r}_t(\alpha'), y_t)$ and $\ell(\alpha', \mathbf{r}_t(\alpha'), y_t) \leq \ell(\alpha', \mathbf{r}_t(\alpha), y_t)$. In other words, by best-responding the agent causes the smallest loss to the learner. Similarly to Case 1, it is easy to see that on these pairs of actions, function $\ell(\alpha, \mathbf{r}_t(\alpha), y_t)$ is again $L_1$-Lipschitz with respect to $\alpha$.

3. For pair $(\alpha, \alpha')$ it holds that

$$\ell(\alpha, \mathbf{r}_t(\alpha), y_t) \geq \ell(\alpha, \mathbf{r}_t(\alpha'), y_t) \tag{4}$$

and

$$\ell(\alpha', \mathbf{r}_t(\alpha'), y_t) \leq \ell(\alpha', \mathbf{r}_t(\alpha), y_t) \tag{5}$$

From Equations (4) and (5) we have that

$$\ell(\alpha', \mathbf{r}_t(\alpha'), y_t) - \ell(\alpha, \mathbf{r}_t(\alpha), y_t) \leq L_1 \|\alpha' - \alpha\| \tag{6}$$

We further distinguish the following cases:

(a) $\ell(\alpha, \mathbf{r}_t(\alpha), y_t) = \ell(\alpha', \mathbf{r}_t(\alpha'), y_t)$. Clearly, $|\ell(\alpha, \mathbf{r}_t(\alpha), y_t) - \ell(\alpha', \mathbf{r}_t(\alpha'), y_t)| \leq L_1 \cdot \|\alpha' - \alpha\|$ holds.

(b) $\ell(\alpha, \mathbf{r}_t(\alpha), y_t) \leq \ell(\alpha', \mathbf{r}_t(\alpha'), y_t)$. From Equation (6), we get: $|\ell(\alpha, \mathbf{r}_t(\alpha), y_t) - \ell(\alpha', \mathbf{r}_t(\alpha'), y_t)| \leq L_1 \cdot \|\alpha' - \alpha\|$.

(c) $\ell(\alpha, \mathbf{r}_t(\alpha), y_t) \geq \ell(\alpha', \mathbf{r}_t(\alpha'), y_t)$ Observe now that if $\ell(\alpha, \mathbf{r}_t(\alpha), y_t) \geq \ell(\alpha', \mathbf{r}_t(\alpha), y_t)$, then from Equation (5) the latter is lower bounded by $\ell(\alpha', \mathbf{r}_t(\alpha'), y_t)$, which leads to a contradiction. Hence, it has to be the case that $\ell(\alpha, \mathbf{r}_t(\alpha), y_t) \leq \ell(\alpha', \mathbf{r}_t(\alpha), y_t)$. The latter, combined with the assumption that $\ell$ is $L_2$ - Lipschitz with respect to its second argument, implies that $\ell(\alpha', \mathbf{r}_t(\alpha'), y_t) - \ell(\alpha, \mathbf{r}_t(\alpha), y_t) \geq -L_2 \cdot \|\mathbf{r}_t(\alpha') - \mathbf{r}_t(\alpha)\|$.

4. For the pair $(\alpha, \alpha')$ it holds that $\ell(\alpha, \mathbf{r}_t(\alpha), y_t) \leq \ell(\alpha, \mathbf{r}_t(\alpha'), y_t)$ and $\ell(\alpha', \mathbf{r}_t(\alpha'), y_t) \geq \ell(\alpha', \mathbf{r}_t(\alpha), y_t)$. The case is analogous to Case 3.

$\blacksquare$

To summarize, in PASGs (Case 1 from aforementioned proof) and PCSGs (Case 2 of aforementioned proof) the loss function written in terms of the action of the agent is *Lipschitz*, i.e., $|\ell(\alpha, \mathbf{r}_t(\alpha), y_t) - \ell(\alpha', \mathbf{r}_t(\alpha'), y_t)| \leq L_1 \cdot \|\alpha' - \alpha\|$. However, in General Stackelberg Games one can only guarantee that

$$|\ell(\alpha, \mathbf{r}_t(\alpha), y_t) - \ell(\alpha', \mathbf{r}_t(\alpha'), y_t)| \leq \max\left\{L_1 \cdot \|\alpha' - \alpha\|, L_2 \cdot \|\mathbf{r}_t(\alpha') - \mathbf{r}_t(\alpha)\|\right\} \tag{7}$$

Using Equation (7), we show that there are some meaningful Stackelberg settings where $\|\mathbf{r}_t(\alpha') - \mathbf{r}_t(\alpha)\|$ can be upper bounded by $\|\alpha' - \alpha\|$ multiplied by a constant. For example, from well known results in convex optimization (for completeness see Lemma A.6), we can see that this is exactly the case in settings where the agent's utility function, $u_t(\alpha, r)$ is *strongly* concave in $r$, and quasilinear[13] in $\alpha$.

**Lemma A.6** (Closeness of Maxima of Strongly Concave Functions (folklore)). *Let functions* $f : \mathcal{X} \mapsto \mathbb{R}, g : \mathcal{X} \mapsto \mathbb{R}$ *be two multidimensional,* $1/\eta_c$-*strongly concave functions with respect to some norm* $\|\cdot\|$. *Let* $h(\mathbf{x}) = f(\mathbf{x}) - g(\mathbf{x}), \mathbf{x} \in \mathcal{X}$ *be* $L_{f,g}$-*Lipschitz[14] with respect to the same norm* $\|\cdot\|$. *Then, for the maxima of the two functions:* $\mu_f = \arg\max_{\mathbf{x} \in \mathcal{X}} f(\mathbf{x})$ *and* $\mu_g = \arg\max_{\mathbf{x} \in \mathcal{X}} g(\mathbf{x})$ *it holds that:*

$$\|\mu_f - \mu_g\| \leq L_{f,g} \cdot \eta_c \tag{8}$$

*Proof.* First, we take the Taylor expansion of $f$ around its maximum, $\mu_f$ and use the strong concavity condition:

$$f(\mathbf{x}) \leq f(\mu_f) + \langle \nabla f(\mu_f), \mathbf{x} - \mu_f \rangle - \frac{1}{2\eta}\|\mu_f - \mathbf{x}\|^2 \qquad \text{(strong concavity)}$$

$$= f(\mu_f) - \frac{1}{2\eta}\|\mu_f - \mathbf{x}\|^2 \qquad (\nabla f(\mu_f) = 0, \text{ since } \mu_f \text{ is the maximum})$$

Similarly, by taking the Taylor expansion of $g$ around its maximum and using the strong concavity condition:

$$g(\mathbf{x}) \leq g(\mu_g) - \frac{1}{2\eta}||\mu_g - \mathbf{x}||^2 \tag{9}$$

Using the $L_{f,g}$-Lipschitzness of $h(\mathbf{x})$ we get:

$$\begin{aligned}
L_{f,g} \cdot ||\mu_g - \mu_f|| \geq |h(\mu_g) - h(\mu_f)| &\geq h(\mu_g) - h(\mu_f) \\
&\geq f(\mu_g) - f(\mu_f) + g(\mu_f) - g(\mu_g) \\
&\geq \frac{1}{2\eta}||\mu_f - \mu_g||^2 + \frac{1}{2\eta}||\mu_f - \mu_g||^2 \quad \text{(from Taylor expansion)} \\
&\geq \frac{1}{\eta}||\mu_f - \mu_g||^2
\end{aligned}$$

Dividing both sides with $||\mu_g - \mu_f||$ concludes the proof. ∎

An example of such a utility function in the context of strategic classification (similar to the family of utility functions used in [17]) is presented below.

**Example.** Let $u_t(\alpha, \mathbf{r}(\alpha), \sigma_t) = \langle \alpha, \mathbf{r}(\alpha) \rangle - (\mathbf{x} - \mathbf{r}(\alpha))^2$. Then, we would like to compute an upper bound on the difference between $||\mathbf{r}(\alpha) - \mathbf{r}(\alpha')||$, where $\mathbf{r}(\alpha) = \arg\max_{\mathbf{z} \in \mathcal{X}; \mathbf{x}} u_t(\alpha, \mathbf{z}, \sigma_t)$ and $\mathbf{r}(\alpha') = \arg\max_{\mathbf{z} \in \mathcal{X}; \mathbf{x}} u_t(\alpha', \mathbf{z}, \sigma_t)$. Following Lemma A.6 we can define functions $f(\mathbf{z}) = u_t(\alpha, \mathbf{z}, \sigma_t)$ and $g(\mathbf{z}) = u_t(\alpha', \mathbf{z}, \sigma_t)$. Now, observe that function $h(\mathbf{z}) = f(\mathbf{z}) - g(\mathbf{z})$ is indeed $||\alpha - \alpha'||$-Lipschitz (i.e., the Lipschitzness constant depends on the specific actions):

$$|f(\mathbf{y}) - g(\mathbf{y}) - f(\mathbf{z}) + g(\mathbf{z})| = |\langle \alpha - \alpha', \mathbf{y} - \mathbf{z} \rangle| \leq ||\alpha - \alpha'|| \cdot ||\mathbf{y} - \mathbf{z}||$$

where the last inequality comes from the Cauchy-Schwartz inequality. Furthermore, observe that both $f(\cdot)$ and $g(\cdot)$ are $\frac{1}{2}$-strongly concave. Therefore, from Lemma A.6 we get that:

$$||\mathbf{r}(\alpha) - \mathbf{r}(\alpha')|| \leq \frac{||\alpha - \alpha'||}{2}$$

# B Appendix for Section 4

## B.1 Notation Reference Tables.

Our model and proof use a lot of notation. For easier reference, we summarize the notation used in our analysis in Tables 2 and 3.

| Variable | Description |
|---|---|
| $d \in \mathbb{N}$ | dimension of the problem |
| $\mathcal{A} \subseteq [-1, 1]^{d+1}$ | learner's action space |
| $\alpha_t \in \mathcal{A}$ | learner's committed action for round $t$ |
| $\mathcal{X} \subseteq ([0, 1]^d, 1)$ | agent's feature vector space |
| $\mathcal{Y} = \{-1, +1\}$ | labels' space |
| $\mathbf{x}_t \in \mathcal{X}$ | agent's feature vector, *as chosen by nature* |
| $\sigma_t = (\mathbf{x}_t, y_t), y_t \in \mathcal{Y}$ | agent's labeled datapoint, *as chosen by nature* |
| $\mathbf{r}_t(\alpha_t, \sigma_t) \in \mathcal{X}$ (simplified to $\mathbf{r}_t(\alpha_t)$) | agent's *reported* feature vector |
| $\hat{y}_t \in \mathcal{Y}$ | $\mathbf{r}_t(\alpha_t)$'s label |
| $\ell(\alpha_t, \mathbf{r}_t(\alpha_t), y_t)$ | learner's loss for action $\alpha_t$ against agent's report $\mathbf{r}_t(\alpha_t)$ |
| $u_t(\alpha_t, \mathbf{r}_t(\alpha_t), \sigma_t)$ | agent's utility for reporting $\mathbf{r}_t(\alpha_t)$, when learner commits to $\alpha_t$ |
| $R(T)$ | learner's *external* regret after $T$ rounds |
| $\mathcal{R}(T)$ | learner's *Stackelberg* regret after $T$ rounds |
| $\lambda(A)$ | Lebesgue measure of measurable space $A$ |

Table 2: Model Notation Summary

| Variable | Description |
|---|---|
| $\mathcal{P}_t$ | set of active polytopes at round $t$ |
| $\overline{\mathcal{P}}_t$ | set of active point-polytopes at round $t$ |
| $\mathcal{D}_t$ | induced distribution from 2-step sampling process |
| $\mathbb{P}_t, f_t$ | cdf and pdf of $\mathcal{D}_t$ |
| $\beta_t^u(\alpha_t) : \langle \mathbf{r}_t(\alpha_t), \mathbf{w} \rangle = 4\sqrt{d}\delta$ | upper boundary hyperplane |
| $\beta_t^l(\alpha_t) : \langle \mathbf{r}_t(\alpha_t), \mathbf{w} \rangle = -4\sqrt{d}\delta$ | lower boundary hyperplane |
| $H^+\left(\beta_t^u(\alpha)\right)$ | $\alpha' \in H^+\left(\beta_t^u(\alpha)\right)$, if $\langle \mathbf{r}_t(\alpha), \alpha' \rangle \geq 4\sqrt{d}\delta$ |
| $H^-\left(\beta_t^l(\alpha)\right)$ | $\alpha' \in H^-\left(\beta_t^l(\alpha)\right)$, if $\langle \mathbf{r}_t(\alpha), \alpha' \rangle \leq -4\sqrt{d}\delta$ |
| $\mathcal{P}_t^u(\alpha)$ | upper polytopes set ($p \in \mathcal{P}_t : p \in H^+\left(\beta_t^u(\alpha)\right)$) |
| $\mathcal{P}_t^l(\alpha)$ | lower polytopes set ($p \in \mathcal{P}_t : p \in H^-\left(\beta_t^l(\alpha)\right)$) |
| $\mathcal{P}_t^m(\alpha)$ | middle polytopes set ($p \in \mathcal{P}_t : p \notin \mathcal{P}_t \setminus \left(\mathcal{P}_t^l \bigcup \mathcal{P}_t^l\right)$ |
| $\mathbb{P}_t^{\text{in}}[\alpha], \mathbb{P}_t^{\text{in}}[p]$ | in-probability for $\alpha$ and $p$ (see Definition 4.2) |
| $\underline{p}_\delta$ | polytope ($\notin \overline{\mathcal{P}}_t$) with smallest Lebesgue measure at round $T$ |

Table 3: Notation Summary for Regret Analysis of GRINDER.

## B.2 Proof of Theorem 4.1.

The proof of Theorem 4.1 follows from a sequence of lemmas and claims presented below. By convention, we call a single point a *point-polytope*, and we denote the set of all point-polytopes by $\overline{\mathcal{P}}$.

**Proposition B.1.** *The two-stage sampling probability distribution $\mathcal{D}_t$ is equivalent to a one-stage probability distribution of drawing directly* an action *from density $d\pi_t(\cdot)$.*

*Proof.* The one-stage probability distribution that draws an action from $\pi_t$ is equivalent to choosing an action $\alpha \in \mathcal{A}$ from probability *density* function: $d\pi_t(\alpha) = (1-\gamma)dq_t(\alpha) + \frac{\gamma}{\lambda(\mathcal{A})}$. The two-stage probability is: $d\pi_{\mathcal{D}_t}(\alpha) = \frac{1}{\lambda(p)}\left((1-\gamma)q_t(p) + \frac{\gamma\lambda(p)}{\lambda(\mathcal{A})}\right)$. Since $q_t(p) = \lambda(p)dq_t(\alpha), \forall \alpha \in p$, we get the result. ∎

Moving forward we analyze the first and the second moment of the loss $\hat{\ell}(\alpha, \mathbf{r}_t(\alpha), y_t)$ for each action $\alpha$, based on the induced probability distribution $\mathcal{D}_t$, assuming oracle access to $\mathbb{P}_t^{\text{in}}[\alpha]$.

**Lemma B.2** (First Moment). *The estimated loss $\hat{\ell}(\alpha, \mathbf{r}_t(\alpha), y_t)$ is an unbiased estimator of the true loss $\ell(\alpha, \mathbf{r}_t(\alpha), y_t)$, when actions are drawn from the induced probability distribution $\mathcal{D}_t$.*

*Proof.* For all the actions $\alpha \in \mathcal{A}$, given Proposition B.1, it holds that:

$$\mathop{\mathbb{E}}_{\alpha_t \sim \mathcal{D}_t}\left[\hat{\ell}(\alpha, \mathbf{r}_t(\alpha), y_t)\right] = \int_{\mathcal{A}} f_t(\alpha') \frac{\ell(\alpha, \mathbf{r}_t(\alpha), y_t)\mathbb{1}\{\alpha \in N^{out}(\alpha')\}}{\mathbb{P}_t^{\text{in}}[\alpha]} d\alpha' = \ell(\alpha, \mathbf{r}_t(\alpha), y_t)$$

∎

**Lemma B.3** (Second Moment). *For the second moment of the estimated loss $\hat{\ell}(\alpha, \mathbf{r}_t(\alpha), y_t)$ with respect to the induced probability distribution $\mathcal{D}_t$ it holds that:*

$$\mathop{\mathbb{E}}_{\alpha_t \sim \mathcal{D}_t}\left[\hat{\ell}(\alpha, \mathbf{r}_t(\alpha), y_t)^2\right] = \frac{\ell(\alpha, \mathbf{r}_t(\alpha), y_t)^2}{\mathbb{P}_t^{\text{in}}[\alpha]} \leq \frac{1}{\mathbb{P}_t^{\text{in}}[\alpha]}$$

*Proof.* For all the actions $\alpha \in \mathcal{A}$, given Claim B.1, it holds that:

$$\mathop{\mathbb{E}}_{\alpha_t \sim \mathcal{D}_t}\left[\hat{\ell}(\alpha, \mathbf{r}_t(\alpha), y_t)^2\right] = \int_{\mathcal{A}} f_t(\alpha') \frac{\ell(\alpha, \mathbf{r}_t(\alpha), y_t)^2\mathbb{1}\{\alpha \in N^{out}(\alpha')\}}{\mathbb{P}_t^{\text{in}}[\alpha]^2} d\alpha' = \frac{\ell(\alpha, \mathbf{r}_t(\alpha), y_t)^2}{\mathbb{P}_t^{\text{in}}[\alpha]} \leq \frac{1}{\mathbb{P}_t^{\text{in}}[\alpha]}$$

∎

**Lemma B.4.** *Let $\underline{p}_\delta(t) = \arg\min_{p \in \mathcal{P}_t \setminus \overline{\mathcal{P}}_t} \lambda(p)$ be the polytope with the smallest Lebesgue measure (excluding point-polytopes) after $t$ rounds. Then, the following inequality holds:*

$$\mathop{\mathbb{E}}_{\alpha_t \sim \mathcal{D}_t} \left[ \frac{1}{\mathbb{P}_t^{\mathtt{in}}[\alpha_t]} \right] \leq 4 \log \left( \frac{4\lambda(\mathcal{A}) \cdot \left| \mathcal{P}_{t,\sigma_t}^u \bigcup \mathcal{P}_{t,\sigma_t}^l \right|}{\gamma \lambda \left( \underline{p}_\delta(t) \right)} \right) + \lambda \left( \mathcal{P}_{t,\sigma_t}^m \right)$$

*Proof.* By definition, we expand the term: $\mathbb{E}_{\alpha \sim \mathcal{D}_t} \left[ \frac{1}{\mathbb{P}_t^{\mathtt{in}}[\alpha_t]} \right]$ as follows:

$$
\begin{aligned}
\mathop{\mathbb{E}}_{\alpha_t \sim \mathcal{D}_t} \left[ \frac{1}{\mathbb{P}_t^{\mathtt{in}}[\alpha_t]} \right] &= \int_{\mathcal{A}} \frac{f_t(\alpha)}{\mathbb{P}_t^{\mathtt{in}}[\alpha]} d\alpha \\
&= \underbrace{\int_{\bigcup \left( \mathcal{P}_{t,\sigma_t}^u \cup \mathcal{P}_{t,\sigma_t}^l \right)} \frac{f_t(\alpha)}{\mathbb{P}_t^{\mathtt{in}}[\alpha]} d\alpha}_{Q_1} + \underbrace{\int_{\bigcup \mathcal{P}_{t,\sigma_t}^m} \frac{f_t(\alpha)}{\mathbb{P}_t^{\mathtt{in}}[\alpha]} d\alpha}_{Q_2}
\end{aligned} \tag{10}
$$

where by integrating over $\bigcup \mathcal{P}$ we denote the integral over all *actions* that belong in some polytope from the set $\mathcal{P}$. In the right hand side of Equation (10), term $Q_2$ is relatively easier to analyze. Due to the conservative estimates of the *true* middle space (i.e., the actions such that $\mathtt{sdist}(\alpha, \mathbf{x}_t) \leq \delta$), the set of polytopes $\mathcal{P}_{t,\sigma_t}^m$ contains *all* the actions that actually belong in the $\sigma_t$-induced middle space, plus some other actions for which the agent could not have misreported, due to their $\delta$-boundedness. Now, for all the actions that actually belong in the $\sigma_t$-induced middle space, it holds that they only get information (i.e., get updated) when they are chosen by the algorithm, while for the rest of the actions that have ended up in our middle space, they could have been updated by other actions as well. Thus, it holds that:

$$\forall \alpha \in \bigcup \mathcal{P}_{t,\sigma_t}^m : \mathbb{P}_t^{\mathtt{in}}[\alpha] \geq f_t(\alpha)$$

As a result:

$$Q_2 = \int_{\bigcup \mathcal{P}_{t,\sigma_t}^m} \frac{f_t(\alpha)}{\mathbb{P}_t^{\mathtt{in}}[\alpha]} d\alpha \leq \int_{\bigcup \mathcal{P}_{t,\sigma_t}^m} \frac{f_t(\alpha)}{f_t(\alpha)} d\alpha = \lambda \left( \mathcal{P}_{t,\sigma_t}^m \right) \tag{11}$$

Moving forward, we turn our attention to term $Q_1$. Assume now that an action $\alpha$ belongs in a polytope $p_\alpha$. Then, there are (weakly) more actions that can potentially update action $\alpha$, than the whole polytope in which it belongs, $p_\alpha$; indeed, in order to update the polytope, one must make sure that every action within it is updateable. As a result, $\mathbb{P}_t^{\mathtt{in}}[\alpha] \geq \mathbb{P}_t^{\mathtt{in}}[p_\alpha]$. Using this in Equation (10) we get that the first term of the RHS of the variance is upper bounded by:

$$Q_1 \leq \sum_{p \in \mathcal{P}_{t,\sigma_t}^u \cup \mathcal{P}_{t,\sigma_t}^l} \int_p \frac{f_t(\alpha)}{\mathbb{P}_t^{\mathtt{in}}[p]} d\alpha \tag{12}$$

Further, let $\mathbb{P}_t^{\mathtt{in}}[p]_{u,l}$ be the part of $\mathbb{P}_t^{\mathtt{in}}[p]$ that depends only in the updates that stem from actions in either the upper or the lower polytopes sets. As such: $\mathbb{P}_t^{\mathtt{in}}[p]_{u,l} \leq \mathbb{P}_t^{\mathtt{in}}[p]$ and the term in Equation (12) can be upper bounded by:

$$Q_1 \leq \sum_{p \in \mathcal{P}_{t,\sigma_t}^u \cup \mathcal{P}_{t,\sigma_t}^l} \frac{1}{\mathbb{P}_t^{\mathtt{in}}[p]_{u,l}} \int_p f_t(\alpha) d\alpha \tag{13}$$

where we have also used the fact that we gain oracle access to quantity $\mathbb{P}_t^{\mathtt{in}}[p]_{u,l}$ and therefore, we treat it as a constant in the integral. Observe now that the term $\int_p f_t(\alpha) d\alpha$ corresponds to the total probability that the action $\alpha_t$, which is chosen from the induced probability distribution $\mathcal{D}_t$, belongs to polytope $p$, i.e., it is equal to $\pi_t(p)$. Hence, the upper bound in Equation (13) can be relaxed to:

$$Q_1 \leq \sum_{p \in \mathcal{P}_{t,\sigma_t}^u \cup \mathcal{P}_{t,\sigma_t}^l} \frac{\pi_t(p)}{\mathbb{P}_t^{\mathtt{in}}[p]_{u,l}} \tag{14}$$

As we have explained before, $\pi_t(p) = 0$, for $p \in \overline{\mathcal{P}}_t$ and as a result, we can disregard point-polytopes from our consideration for the rest of this proof. We now upper bound this term by using the graph-theoretic lemma of Alon et al. [1, Lemma 5], which we provide below for completeness.

**Lemma B.5** ([1, Lemma 5]). *Let $G = (V, E)$ be a directed graph with $|V| = K$, in which each node $i \in V$ is assigned a positive weight $w_i$ lower bounded by a positive scalar $\varepsilon \in (0, 1/2)$, i.e., $w_i \geq \varepsilon, \forall i \in V$. If $\sum_{i \in V} w_i \leq 1$ then, denoting by $\alpha^G$ the independence number of $G$ we have that:*

$$\sum_{i \in V} \frac{w_i}{w_i + \sum_{j \in N^{in}(i)} w_j} \leq 4\alpha^G \frac{4K}{\alpha^G \varepsilon}$$

Observe that all the actions within the $\sigma_t$-induced upper and the lower polytopes set form the following feedback graph: each node corresponds to a polytope from one of the sets $\mathcal{P}^u_{t,\sigma_t}, \mathcal{P}^l_{t,\sigma_t}$. So the total number of nodes is at most $\left|\mathcal{P}^u_{t,\sigma_t} \cup \mathcal{P}^l_{t,\sigma_t}\right|$, where by $|S|$ we denote the cardinality of a set $S$. Each edge $(i, j)$ corresponds to *information passing* from node $i$ to node $j$, i.e., the directed edge $(i, j)$ exists when the loss for actions of polytope $j$ can be computed by just observing the loss for action from the polytope $i$. However, for each action belonging in a polytope among the $\sigma_t$-induced upper and lower polytopes sets, we know that the agent could not possibly misreport, due to him being myopically rational and $\delta$-bounded, and as a result, the loss for all the actions within the upper and the lower polytopes sets can be computed! As a result, the independence number of this feedback graph is $\alpha^G = 1$. Using the fact that each polytope $p$ is chosen with probability at least $\pi_t(p) \geq \gamma \frac{\lambda(p)}{\lambda(\mathcal{A})} \geq \gamma \frac{\lambda(\underline{p}_\delta(t))}{\lambda(\mathcal{A})}$ we can apply Lemma B.5 for $\varepsilon = \gamma \frac{\lambda(\underline{p}_\delta(t))}{\lambda(\mathcal{A})}$ and $\alpha^G = 1$ and obtain:

$$Q_1 \leq 4 \log \left( \frac{4\lambda(\mathcal{A}) \cdot \left|\mathcal{P}^u_{t,\sigma_t} \cup \mathcal{P}^l_{t,\sigma_t}\right|}{\lambda\left(\underline{p}_\delta(t)\right) \cdot \gamma} \right)$$

Summing up the upper bounds for $Q_1$ and $Q_2$ we get:

$$\mathbb{E}_{\alpha_t \sim \mathcal{D}_t} \left[ \frac{1}{\mathbb{P}^{in}_t[\alpha_t]} \right] \leq 4 \log \left( \frac{4\lambda(\mathcal{A}) \cdot \left|\mathcal{P}^u_{t,\sigma_t} \cup \mathcal{P}^l_{t,\sigma_t}\right|}{\lambda\left(\underline{p}_\delta(t)\right) \cdot \gamma} \right) + \lambda\left(\mathcal{P}^m_{t,\sigma_t}\right)$$

∎

**Lemma B.6** (Second Order Regret Bound). *Let $q_1, \ldots, q_T$ be the probability distribution over the polytopes defined by in Step 15 of Algorithm 2 for the estimated losses $\hat{\ell}(\alpha, \mathbf{r}_t(\alpha), y_t), t \in [T]$. Then, the second order regret bound induced by GRINDER is:*

$$\sum_{t=1}^{T} \sum_{p \in \mathcal{P}_{t+1}} q_t(p)\hat{\ell}(p, \mathbf{r}_t(p), y_t) - \sum_{t=1}^{T} \hat{\ell}(\alpha^\star, \mathbf{r}_t(\alpha^\star), y_t) \leq \frac{\eta}{2} \sum_{t=1}^{T} \sum_{p \in \mathcal{P}_{t+1}} q_t(p)\hat{\ell}(p, \mathbf{r}_t(p), y_t)^2 + \frac{1}{\eta} \log \left( \frac{\lambda(\mathcal{A})}{\lambda(\underline{p})} \right)$$
(15)

*where $\underline{p}_\delta$ is the polytope with the smallest Lebesgue measure in the finest partition of space $\mathcal{A}$: $\underline{p}_\delta = \arg\min_{p \in \mathcal{P}_T \setminus \overline{\mathcal{P}}_T} \lambda(p)$.*

*Proof.* Let $W_t = \sum_{p \in \mathcal{P}_t} w_t(p)$. We upper and lower bound the quantity $Q = \sum_{t=1}^{T} \log \frac{W_{t+1}}{W_t}$. For the lower bound:

$$Q = \sum_{t=1}^{T} \log \left( \frac{W_{t+1}}{W_t} \right) = \log \left( \frac{W_T}{W_1} \right)$$
(16)

Observe now that in $t = 1$ there only exists one polytope (the whole $[-1, 1]^{d+1}$ space), with a total weight of $\lambda(\mathcal{A})$ and a probability of 1. In other words, all the actions within this polytope have the same weight, which is equal to 1 (uniformly weighted). As a result, $\log W_1 = \log \left( \sum_{p \in \mathcal{P}_1} \int_{\mathcal{A}} 1 d\alpha \right) = \log(\lambda(\mathcal{A}))$. For term $\log W_T$ we have:

$$\log W_T = \log \left( \sum_{p \in \mathcal{P}_T} w_T(p) \right) = \log \left( \int_{\mathcal{A}} w_T(\alpha) d\alpha \right)$$

$$= \log \left( \sum_{p \in \mathcal{P}_T \setminus \overline{\mathcal{P}}_T} \lambda(p) \exp \left( -\eta \sum_{t=1}^{T} \hat{\ell}(p, \mathbf{r}_t(p), y_t) \right) + \int_{\cup \overline{\mathcal{P}}_T} \exp \left( -\eta \sum_{t=1}^{T} \hat{\ell}(\alpha, \mathbf{r}_t(\alpha), y_t) \right) d\alpha \right)$$
(17)

where the last equality is due to the fact that not further grinded polytopes have maintained the *same* estimated loss, $\hat{\ell}$, for *all* their containing points at each round $t$ and we denote by $\overline{\mathcal{P}}_T$ the set of point-polytopes contained in $\mathcal{P}_T$.

Since the horizon $T$ is finite, set $\overline{\mathcal{P}}_T$ is essentially a set of points, and it has a Lebesgue measure of 0. Hence,

$$\int_{\bigcup \overline{\mathcal{P}}_T} \exp\left(-\eta \sum_{t=1}^{T} \hat{\ell}(\alpha, \mathbf{r}_t(\alpha), y_t)\right) d\alpha = 0$$

Let $\alpha^\star = \arg\min_{\alpha \in \mathcal{A}} \sum_{t=1}^{T} \hat{\ell}(\alpha, \mathbf{r}_t(\alpha), y_t)$ (i.e., the best fixed action in hindsight among the *all* actions after $T$ rounds, irrespective of whether it belongs to $\bigcup \overline{\mathcal{P}}_T$ or $\bigcup \mathcal{P}_T \setminus \overline{\mathcal{P}}_T$) and $\underline{p}_\delta \in \mathcal{P}_T \setminus \overline{\mathcal{P}}_T$ be the polytope with the smallest Lebesgue measure in $\mathcal{P}_T \setminus \overline{\mathcal{P}}_T$ (i.e., excluding point-polytopes). Then, denoting by $p^* \in \mathcal{P}_T$ the polytope where $\alpha^\star$ belongs to, among the set of active polytopes $\mathcal{P}_T$, Equation (17) becomes can be lower bounded as follows:

$$
\begin{aligned}
\log W_T &= \log\left(\sum_{p \in \mathcal{P}_T \setminus \overline{\mathcal{P}}_T} \lambda(p) \exp\left(-\eta \sum_{t=1}^{T} \hat{\ell}(p, \mathbf{r}_t(p), y_t)\right)\right) \\
&\geq \log\left(\lambda(\underline{p}_\delta) \sum_{p \in \mathcal{P}_T \setminus \overline{\mathcal{P}}_T} \exp\left(-\eta \sum_{t=1}^{T} \hat{\ell}(p, \mathbf{r}_t(p), y_t)\right)\right) \\
&\qquad\qquad\qquad\qquad\qquad\qquad\qquad (\lambda(p) \geq \lambda(\underline{p}_\delta), \forall p \in \mathcal{P}_T \setminus \overline{\mathcal{P}}_T) \\
&\geq \log\left(\lambda(\underline{p}_\delta) \cdot \exp\left(-\eta \sum_{t=1}^{T} \hat{\ell}(p^*, \mathbf{r}_t(p^*), y_t)\right)\right) \qquad\qquad (e^{-x} \geq 0, \forall x) \\
&= \log\left(\lambda\left(\underline{p}_\delta\right) \cdot \exp\left(-\eta \sum_{t=1}^{T} \hat{\ell}(\alpha^\star, \mathbf{r}_t(\alpha^\star), y_t)\right)\right) = \log\left(\lambda\left(\underline{p}_\delta\right)\right) - \eta \sum_{t=1}^{T} \hat{\ell}(\alpha^\star, \mathbf{r}_t(\alpha^\star), y_t)
\end{aligned}
$$
$$\tag{18}$$

As a result:

$$Q = \log W_T - \log W_1 \geq \log\left(\frac{\lambda\left(\underline{p}_\delta\right)}{\lambda(\mathcal{A})}\right) - \eta \sum_{t=1}^{T} \hat{\ell}(\alpha^\star, \mathbf{r}_t(\alpha^\star), y_t) \tag{19}$$

We move on to the upper bound of $Q$ now. Upper bounding quantity $\log \frac{W_{t+1}}{W_t}$ we get:

$$
\begin{aligned}
\log\left(\frac{W_{t+1}}{W_t}\right) &= \log\left(\frac{\int_{\mathcal{A}} w_t(\alpha) \exp\left(-\eta \hat{\ell}(\alpha, \mathbf{r}_t(\alpha), y_t)\right) d\alpha}{W_t}\right) \\
&= \log\left(\int_{\mathcal{A}} q_t(\alpha) \exp\left(-\eta \hat{\ell}(\alpha, \mathbf{r}_t(\alpha), y_t)\right) d\alpha\right) \\
&\leq \log\left(\int_{\mathcal{A}} q_t(\alpha)\left(1 - \eta \hat{\ell}(\alpha, \mathbf{r}_t(\alpha), y_t) + \frac{\eta^2}{2} \hat{\ell}(\alpha, \mathbf{r}_t(\alpha), y_t)^2\right) d\alpha\right) \\
&\qquad\qquad\qquad\qquad\qquad\qquad (e^{-x} \leq 1 - x + \frac{x^2}{2}, x \in [0, 1]) \\
&\leq \log\left(1 - \eta \int_{\mathcal{A}} q_t(\alpha)\hat{\ell}(\alpha, \mathbf{r}_t(\alpha), y_t) d\alpha + \frac{\eta^2}{2} \int_{\mathcal{A}} q_t(\alpha)\hat{\ell}(\alpha, \mathbf{r}_t(\alpha), y_t)^2 d\alpha\right) \\
&\qquad\qquad\qquad\qquad\qquad\qquad\qquad\qquad (\int_{\mathcal{A}} q_t(\alpha)d\alpha = 1) \\
&\leq -\eta \int_{\mathcal{A}} q_t(\alpha)\hat{\ell}(\alpha, \mathbf{r}_t(\alpha), y_t) d\alpha + \frac{\eta^2}{2} \int_{\mathcal{A}} q_t(\alpha)\hat{\ell}(\alpha, \mathbf{r}_t(\alpha), y_t)^2 d\alpha \quad (\log(1-x) \leq x, x \leq 0)
\end{aligned}
$$

Summing up for the $T$ rounds the latter becomes:

$$\sum_{t=1}^{T} \log\left(\frac{W_{t+1}}{W_t}\right) \leq -\sum_{t=1}^{T} \eta \int_{\mathcal{A}} q_t(\alpha)\hat{\ell}(\alpha, \mathbf{r}_t(\alpha), y_t) d\alpha + \sum_{t=1}^{T} \frac{\eta^2}{2} \int_{\mathcal{A}} q_t(\alpha)\hat{\ell}(\alpha, \mathbf{r}_t(\alpha), y_t)^2 d\alpha$$
$$\tag{20}$$

Combining the upper and lower bounds of Equations (19) and (20) we get that:

$$\sum_{t=1}^{T} \int_{\mathcal{A}} q_t(\alpha)\hat{\ell}(\alpha, \mathbf{r}_t(\alpha), y_t) d\alpha - \sum_{t=1}^{T} \hat{\ell}(\alpha^\star, \mathbf{r}_t(\alpha^\star), y_t) \leq \frac{\eta}{2} \sum_{t=1}^{T} \int_{\mathcal{A}} q_t(\alpha)\hat{\ell}(\alpha, \mathbf{r}_t(\alpha), y_t)^2 d\alpha + \frac{1}{\eta} \log\left(\frac{\lambda(\mathcal{A})}{\lambda\left(\underline{p}_\delta\right)}\right)$$

∎

We are now ready for the proof of Theorem 4.1.

*Proof of Theorem 4.1.* By taking the expectation with respect to distribution $\mathcal{D}_t$ in Lemma B.6 we get that:

$$\sum_{t=1}^{T} \int_{\mathcal{A}} q_t(\alpha) \mathop{\mathbb{E}}_{\mathcal{D}_t}\left[\hat{\ell}(\alpha, \mathbf{r}_t(\alpha), y_t)\right] d\alpha - \sum_{t=1}^{T} \mathop{\mathbb{E}}_{\mathcal{D}_t}\left[\hat{\ell}(\alpha^\star, \mathbf{r}_t(\alpha^\star), y_t)\right] \leq$$

$$\leq \frac{\eta}{2} \sum_{t=1}^{T} \int_{\mathcal{A}} q_t(\alpha) \mathop{\mathbb{E}}_{\mathcal{D}_t}\left[\hat{\ell}(\alpha, \mathbf{r}_t(\alpha), y_t)^2\right] d\alpha + \frac{1}{\eta} \log\left(\frac{\lambda(\mathcal{A})}{\lambda\left(\underline{p}_\delta\right)}\right)$$

Combining Lemmas B.2, B.3 with the latter we get:

$$\sum_{t=1}^{T} \int_{\mathcal{A}} q_t(\alpha)\ell(\alpha, \mathbf{r}_t(\alpha), y_t) d\alpha - \sum_{t=1}^{T} \ell(\alpha^\star, \mathbf{r}_t(\alpha^\star), y_t)$$

$$\leq \sum_{t=1}^{T} \frac{\eta}{2} \int_{\mathcal{A}} \frac{q_t(\alpha)}{\mathbb{P}_t^{\text{in}}[\alpha]} d\alpha + \frac{1}{\eta} \log\left(\frac{\lambda(\mathcal{A})}{\lambda\left(\underline{p}_\delta\right)}\right)$$

$$\leq \sum_{t=1}^{T} \eta \int_{\mathcal{A}} \frac{\pi_t(\alpha)}{\mathbb{P}_t^{\text{in}}[\alpha]} d\alpha + \frac{1}{\eta} \log\left(\frac{\lambda(\mathcal{A})}{\lambda\left(\underline{p}_\delta\right)}\right) \qquad (\pi_t(\alpha) \geq (1-\gamma)q_t(\alpha) \text{ and } \gamma \leq \tfrac{1}{2})$$

$$\leq \sum_{t=1}^{T} \eta\left(4\log\left(\frac{4\lambda(\mathcal{A})\left|\mathcal{P}_{t,\sigma_t}^u \bigcup \mathcal{P}_{t,\sigma_t}^l\right|}{\gamma \cdot \lambda\left(\underline{p}_\delta(t)\right)}\right) + \lambda\left(\mathcal{P}_{t,\sigma_t}^m\right)\right) + \frac{1}{\eta} \log\left(\frac{\lambda(\mathcal{A})}{\lambda\left(\underline{p}_\delta\right)}\right)$$
$$\text{(Lemma B.4)}$$

Using the fact that $\int_{\mathcal{A}} \pi_t(\alpha) d\alpha \leq \int_{\mathcal{A}} q_t(\alpha) d\alpha + \gamma$, the latter becomes:

$$\mathcal{R}(T) \leq \gamma T + \eta \sum_{t=1}^{T}\left(4\log\left(\frac{4\lambda(\mathcal{A})\left|\mathcal{P}_{t,\sigma_t}^u \bigcup \mathcal{P}_{t,\sigma_t}^l\right|}{\gamma \cdot \lambda\left(\underline{p}_\delta(t)\right)}\right) + \lambda\left(\mathcal{P}_{t,\sigma_t}^m\right)\right) + \frac{1}{\eta} \log\left(\frac{\lambda(\mathcal{A})}{\lambda\left(\underline{p}_\delta\right)}\right)$$

Setting $\gamma = \eta$:

$$\mathcal{R}(T) \leq \eta \sum_{t=1}^{T}\left(1 + 4\log\left(\frac{4\lambda(\mathcal{A})\left|\mathcal{P}_{t,\sigma_t}^u \bigcup \mathcal{P}_{t,\sigma_t}^l\right|}{\eta \cdot \lambda\left(\underline{p}_\delta(t)\right)}\right) + \lambda\left(\mathcal{P}_{t,\sigma_t}^m\right)\right) + \frac{1}{\eta} \log\left(\frac{\lambda(\mathcal{A})}{\lambda\left(\underline{p}_\delta\right)}\right)$$

which can be relaxed to:

$$\mathcal{R}(T) \leq \eta \sum_{t=1}^{T}\left(1 + 4\log\left(\frac{4\lambda(\mathcal{A})\left|\mathcal{P}_{t,\sigma_t}^u \bigcup \mathcal{P}_{t,\sigma_t}^l\right|}{\lambda\left(\underline{p}_\delta(t)\right)}\right) + \lambda\left(\mathcal{P}_{t,\sigma_t}^m\right)\right) + \frac{1}{\eta} \log\left(\frac{\lambda(\mathcal{A})}{\lambda\left(\underline{p}_\delta\right)}\right)$$

$$\leq \eta \sum_{t=1}^{T}\left(1 + 4\log\left(\frac{4\lambda(\mathcal{A})\left|\mathcal{P}_{t,\sigma_t}^u \bigcup \mathcal{P}_{t,\sigma_t}^l\right|T}{\lambda\left(\underline{p}_\delta\right)}\right) + \lambda\left(\mathcal{P}_{t,\sigma_t}^m\right)\right) + \frac{1}{\eta} \log\left(\frac{\lambda(\mathcal{A})}{\lambda\left(\underline{p}_\delta\right)}\right)$$
$$(\lambda(\underline{p}_\delta(t)) \geq \lambda(\underline{p}_\delta))$$

$$\leq \eta \cdot \max_{t \in [T]}\left\{1 + 4\log\left(\frac{4\lambda(\mathcal{A})\left|\mathcal{P}_{t,\sigma_t}^u \bigcup \mathcal{P}_{t,\sigma_t}^l\right|T}{\lambda\left(\underline{p}_\delta\right)}\right) + \lambda\left(\mathcal{P}_{t,\sigma_t}^m\right)\right\} \cdot T + \frac{1}{\eta} \log\left(\frac{\lambda(\mathcal{A})}{\lambda\left(\underline{p}_\delta\right)}\right)$$

Tuning $\eta$ to be

$$\eta = \sqrt{\frac{\log\left(\frac{\lambda(\mathcal{A})}{\lambda(\underline{p}_\delta)}\right)}{\max_{t\in[T]}\left\{1 + 4\log\left(\frac{4\lambda(\mathcal{A})|\mathcal{P}^u_{t,\sigma_t}\cup\mathcal{P}^l_{t,\sigma_t}|T}{\lambda(\underline{p}_\delta)}\right) + \lambda\left(\mathcal{P}^m_{t,\sigma_t}\right)\right\}\cdot T}}$$

we get that the Stackelberg regret is upper bounded by:

$$\mathcal{R}(T) \leq \mathcal{O}\left(\sqrt{\max_{t\in[T]}\left\{\lambda\left(\mathcal{P}^m_{t,\sigma_t}\right) + 4\log\left(\frac{4\lambda(\mathcal{A})\left|\mathcal{P}^u_{t,\sigma_t}\cup\mathcal{P}^l_{t,\sigma_t}\right|T}{\lambda(\underline{p}_\delta)}\right) + 1\right\}\cdot \log\left(\frac{\lambda(\mathcal{A})}{\lambda\left(\underline{p}_\delta\right)}\right)\cdot T}\right)$$

Since the actions that belong in $\mathcal{P}^m_{t,\sigma_t}$ are a subset of all the actions in $\mathcal{A}$, then $\lambda\left(\mathcal{P}^m_{t,\sigma_t}\right) \leq \lambda(\mathcal{A}) = 1$. The set of all polytopes is upper bounded by $\frac{\lambda(\mathcal{A})}{\lambda(\underline{p}_\delta)}$ and hence, $\left|\mathcal{P}^u_{t,\sigma_t}\cup\mathcal{P}^l_{t,\sigma_t}\right| \leq \frac{\lambda(\mathcal{A})}{\lambda(\underline{p}_\delta)}$. Hence, for the Stackelberg regret we have:

$$\mathcal{R}(T) \leq \mathcal{O}\left(\sqrt{\log\left(\frac{\lambda(\mathcal{A})}{\lambda(\underline{p}_\delta)}T\right)\cdot\log\left(\frac{\lambda(\mathcal{A})}{\lambda(\underline{p}_\delta)}\right)T}\right)$$

$$\leq \mathcal{O}\left(\sqrt{\left(\lambda(\mathcal{A}) + 1 + 4\log\left(\frac{4\lambda(\mathcal{A})}{\lambda(\underline{p}_\delta)}\cdot\frac{\lambda(\mathcal{A})}{\lambda(\underline{p}_\delta)}\cdot T\right)\right)\cdot\log\left(\frac{\lambda(\mathcal{A})}{\lambda(\underline{p}_\delta)}\right)T}\right)$$

$$\leq \mathcal{O}\left(\sqrt{\left(\lambda(\mathcal{A}) + 1 + 8\log\left(\frac{2\lambda(\mathcal{A})}{\lambda(\underline{p}_\delta)}\cdot T\right)\right)\cdot\log\left(\frac{\lambda(\mathcal{A})}{\lambda(\underline{p}_\delta)}\right)T}\right)$$

$$\leq \mathcal{O}\left(\sqrt{\log\left(\frac{\lambda(\mathcal{A})}{\lambda(\underline{p}_\delta)}T\right)\cdot\log\left(\frac{\lambda(\mathcal{A})}{\lambda(\underline{p}_\delta)}\right)T}\right)$$

where the $\mathcal{O}(\cdot)$ notation hides constants with respect to the horizon $T$. ∎

### B.3 Remaining Proofs

**Lemma B.7.** *For $\varepsilon \leq 1/2$, we call $\widetilde{\mathbb{P}}_t[\alpha_t]$ an $\varepsilon$-approximation oracle to $\mathbb{P}^{in}_t[\alpha_t]$, if $|\widetilde{\mathbb{P}}_t[\alpha_t] - \mathbb{P}^{in}_t[\alpha_t]| \leq \varepsilon\mathbb{P}^{in}_t[\alpha_t]$. Then, GRINDER run with oracle $\widetilde{\mathbb{P}}_t[\cdot]$ instead of $\mathbb{P}^{in}_t[\alpha_t]$ achieves Stackelberg regret $\mathcal{R}(T) \leq \mathcal{O}(\sqrt{T\log(T\lambda(\mathcal{A})/\lambda(\underline{p}_\delta))\cdot\log(\lambda(\mathcal{A})/\lambda(\underline{p}_\delta))}) + 2\varepsilon T$.*

*Proof of Lemma B.7.* We start by computing how the first moment of estimator $\hat{\ell}$ changes once you reweigh with $\widetilde{\mathbb{P}}_t[\cdot]$ rather than $\mathbb{P}^{in}_t[\cdot]$:

$$\mathbb{E}_{\alpha_t\sim\mathcal{D}_t}\left[\hat{\ell}(\alpha,\mathbf{r}_t(\alpha),y_t)\right] = \int_{\mathcal{A}}f_t\left(\alpha'\right)\frac{\ell(\alpha,\mathbf{r}_t(\alpha),y_t)\mathbb{1}\left\{\alpha\in N^{out}(\alpha')\right\}}{\widetilde{\mathbb{P}}_t[\alpha]}d\alpha'$$

$$= \ell(\alpha,\mathbf{r}_t(\alpha),y_t)\cdot\frac{\mathbb{P}^{in}_t[\alpha]}{\widetilde{\mathbb{P}}_t[\alpha]} \tag{21}$$

Since $\widetilde{\mathbb{P}}_t[\alpha] \geq (1-\varepsilon)\mathbb{P}^{in}_t[\alpha]$, then from Equation (21) we have that:

$$\mathbb{E}_{\alpha_t\sim\mathcal{D}_t}\left[\hat{\ell}(\alpha,\mathbf{r}_t(\alpha),y_t)\right] \leq \frac{\ell(\alpha,\mathbf{r}_t(\alpha),y_t)}{1-\varepsilon} \tag{22}$$

Additionally, since $\widetilde{\mathbb{P}}_t[\alpha] \leq (1+\varepsilon)\mathbb{P}^{in}_t[\alpha]$, then from Equation (21) we have that:

$$\mathbb{E}_{\alpha_t\sim\mathcal{D}_t}\left[\hat{\ell}(\alpha,\mathbf{r}_t(\alpha),y_t)\right] \geq \frac{\ell(\alpha,\mathbf{r}_t(\alpha),y_t)}{1+\varepsilon} \tag{23}$$

We turn our attention to the second moment now, for which we will only need an upper bound.

$$\mathop{\mathbb{E}}_{\alpha_t \sim \mathcal{D}_t} \left[ \hat{\ell}(\alpha, \mathbf{r}_t(\alpha), y_t)^2 \right] = \int_{\mathcal{A}} f_t(\alpha') \frac{\ell(\alpha, \mathbf{r}_t(\alpha), y_t)^2 \mathbb{1}\{\alpha \in N^{out}(\alpha')\}}{\widetilde{\mathbb{P}}_t[\alpha]^2} d\alpha' = \frac{\ell(\alpha, \mathbf{r}_t(\alpha), y_t)^2 \mathbb{P}_t^{\mathtt{in}}[\alpha]}{\widetilde{\mathbb{P}}_t[\alpha]^2}$$

$$\leq \frac{1}{(1-\varepsilon)^2 \mathbb{P}_t^{\mathtt{in}}[\alpha]} \tag{24}$$

Lemma B.4 still holds without any change, as it is not affected by the exact definition of $\hat{\ell}(\cdot)$, and so does Lemma B.6. Taking expectations in Lemma B.6 we obtain the following:

$$\sum_{t=1}^{T} \sum_{p \in \mathcal{P}_{t+1}} q_t(p) \mathbb{E}\left[ \hat{\ell}(p, \mathbf{r}_t(p), y_t) \right] - \sum_{t=1}^{T} \mathbb{E}\left[ \hat{\ell}(\alpha^\star, \mathbf{r}_t(\alpha^\star), y_t) \right]$$

$$\leq \frac{\eta}{2} \sum_{t=1}^{T} \sum_{p \in \mathcal{P}_{t+1}} q_t(p) \mathbb{E}\left[ \hat{\ell}(p, \mathbf{r}_t(p), y_t)^2 \right] + \frac{1}{\eta} \log\left( \frac{\lambda(\mathcal{A})}{\lambda(\underline{p})} \right)$$

Applying Equations (22), (23) and (24) on the latter we obtain:

$$\frac{1}{1+\varepsilon} \sum_{t=1}^{T} \int_{\mathcal{A}} q_t(\alpha) \ell(\alpha, \mathbf{r}_t(\alpha), y_t) d\alpha - \frac{1}{1-\varepsilon} \sum_{t=1}^{T} \ell(\alpha^\star, \mathbf{r}_t(\alpha^\star), y_t)$$

$$\leq \frac{\eta}{2} \frac{1}{(1-\varepsilon)^2} \sum_{t=1}^{T} \int_{\mathcal{A}} \frac{q_t(\alpha) d\alpha}{\mathbb{P}_t^{\mathtt{in}}[\alpha]} + \frac{1}{\eta} \log\left( \frac{\lambda(\mathcal{A})}{\lambda(\underline{p})} \right)$$

In the latter, applying Lemma B.4, multiplying both sides by $1 - \varepsilon$ and using the fact that $\varepsilon \leq 1/2$ we obtain the result. ∎

**Lemma B.8.** *Provided access to algorithms for computing the volume of a polytope and to an in-probability oracle,* GRINDER *has runtime complexity* $\mathcal{O}(T^d)$.

*Proof of Lemma B.8.* With access to algorithms that compute the volume of a polytope and to an in-probability oracle, the complexity of GRINDER is dependent solely on the number of polytopes that get activated in the worst case. The latter depends on the number of new boundary hyperplanes that we introduce in the action space $\mathcal{A}$ at each round.

If the sequence of real feature vectors $\{\mathbf{x}_t\}_{t=1}^{T}$ is chosen adversarially, the number of new hyperplanes added in each round in $\mathcal{A}$ is 2. So, *in the worst case*, after $T$ rounds we have $2T$ hyperplanes in general position in a $d$-dimensional space, which from Zaslavsky [34], Stanley et al. [33] are:

$$|\mathcal{P}_t| = O\left( \sum_{i=0}^{d} \binom{2T}{i} \right) = O\left( \frac{T^d}{d!} \right)$$

∎

# C   Appendix for Section 5

**Lemma C.1.** *Fix a* $\mathbf{r} = \mathbf{x} = (u)^d$, *where by* $(u)^d$ *we denote the d-dimensional vector with* $u \in [1/4, 3/4]$ *in every dimension. There exists a utility model for the agents, and a pair of adversarial environments U and L such that* $\mathbf{r}_t(\alpha) = \mathbf{x}_t = \mathbf{x}, \forall \alpha \in \mathcal{A}, \forall t \in [T]$, *and the sequence of* $y_1, \ldots, y_T$ *is i.i.d. conditional on the choice of the adversary, such that:*

$$\max_{\nu \in \{U, L\}} \min_{\alpha^\star \in \mathcal{A}} \mathbb{E}_\nu \left[ \sum_{t \in [T]} \ell(\alpha_t, \mathbf{r}_t(\alpha_t), y_t) - \sum_{t \in [T]} \ell(\alpha^\star, \mathbf{r}_t(\alpha^\star), y_t) \right] \geq \frac{1}{9\sqrt{2}} \sqrt{T}$$

*Proof.* We are going to show this for the case where the agents $\forall t \in [T]$ are *truthful*, i.e., they decide to report $\mathbf{r}_t(\alpha) = \mathbf{x}_t, \forall \alpha \in \mathcal{A}, \forall t \in [T]$. Of course, the learner does not know (and cannot infer) that, so fix a $\delta > 0$ for the $\delta$-boundedness of the agents' utility function. We will prove the lemma

only for deterministic strategies for the learner. As is customary, the claim for general strategies can be concluded by averaging over the learner's internal randomness and Fubini's theorem.

Fix an $\varepsilon > 0$, and a scalar $u \in [1/4, 3/4]$, and define the adversarial environments as follows: $U$ is such that $y_t = +1$ with probability $1/2 + \varepsilon$ and $y_t = -1$ with probability $1/2 - \varepsilon$, and $L$ is such that $y_t = -1$ with probability $1/2 + \varepsilon$ and $y_t = +1$ with probability $1/2 - \varepsilon$. This means that under $U$, the majority of times the label is $+1$, and under $L$, the majority of times the label is $-1$. As a result, under $U$, *any* action $\alpha$ such that $\langle \alpha, \mathbf{x} \rangle \geq 2\delta$ is *optimal* and under $L$, *any* action $\alpha$ such that $\langle \alpha, \mathbf{x} \rangle \leq -2\delta$ is *optimal*.

Take a sequence of actions $\alpha_1, \ldots, \alpha_T$ and let $T_{\geq \delta}$ denote the *number* of rounds for which $\langle \alpha_t, \mathbf{r} \rangle \geq \sqrt{d}\delta$, and $T_{\leq -\delta}$ the number of rounds for which $\langle \alpha_t, \mathbf{r} \rangle \leq -\sqrt{d}\delta$. Since $T_{\leq -\delta} + T_{\geq \delta} \leq T$ we get that:

$$\underset{U}{\mathbb{E}}\left[\mathcal{R}(T)\right] \geq \underset{U}{\mathbb{E}}\left[\mathcal{R}(T_{\leq -\delta})\right]$$

$$\geq \sum_{t \in [T_{\leq -\delta}]} \left[ 1 \cdot \left(\frac{1}{2} + \varepsilon\right) - 1 \cdot \left(\frac{1}{2} - \varepsilon\right) \right]$$

$$\geq 2\varepsilon \underset{U}{\mathbb{E}}\left[T_{\leq -\delta}\right] \tag{25}$$

where the first inequality is due to the fact that $\ell(\alpha_t, \mathbf{x}, y_t) = 0 = \ell(\alpha_U^*, \mathbf{x}, y_t), \forall t \in [T_{\geq \delta}]$ and any optimal action $\alpha_U^*$ under $U$ as we reasoned before. The second inequality uses the following two facts: first, that $\ell(\alpha_U^*, \mathbf{x}, y_t) = 1, \forall t \in [T_{\leq -\delta}]$, i.e., the best fixed action in hindsight when one encounters adversarial environment $U$ is an action that estimates the label of $\mathbf{x}$ to be 1. Second, that when playing against environment $U$, a learner incurs loss of 1 every time that she predicted the label of $\mathbf{x}$ to be $-1$ (which happens in at least all $T_{\leq -\delta}$ rounds), and the actual label was 1 (which happens with probability $1/2 + \varepsilon$). Similarly, we also see that

$$\underset{L}{\mathbb{E}}\left[\mathcal{R}(T)\right] \geq 2\varepsilon \underset{L}{\mathbb{E}}\left[T_{\geq \delta}\right] \tag{26}$$

Let $\mathbb{P}_U, \mathbb{P}_L$ the distributions of $T_{\leq -\delta}, T_{\geq \delta}$ for adversarial environments $U, L$ respectively, and let $\mathbb{P}_m$ be the distribution of rounds when $y_t = +1$ with probability $1/2$. From Pinsker's inequality, and denoting by $\mathtt{KL}(p,q)$ the KL-divergence between distributions $p, q$, we have the following:

$$\underset{U}{\mathbb{E}}\left[T_{\leq -\delta}\right] \geq \underset{m}{\mathbb{E}}\left[T_{\leq -\delta}\right] - T\sqrt{\frac{\mathtt{KL}(\mathbb{P}_U, \mathbb{P}_m)}{2}} \tag{27}$$

and

$$\underset{L}{\mathbb{E}}\left[T_{\geq \delta}\right] \geq \underset{m}{\mathbb{E}}\left[T_{\geq \delta}\right] - T\sqrt{\frac{\mathtt{KL}(\mathbb{P}_U, \mathbb{P}_m)}{2}} \tag{28}$$

Then, from the data processing inequality for the KL-divergence we get:

$$\mathtt{KL}\left(\underset{U}{\mathbb{P}}, \underset{m}{\mathbb{P}}\right) \leq T\mathtt{KL}\left(\mathtt{Bern}\left(\frac{1}{2} + \varepsilon\right), \mathtt{Bern}\left(\frac{1}{2}\right)\right) \leq 4T\varepsilon^2 \tag{29}$$

and

$$\mathtt{KL}\left(\underset{L}{\mathbb{P}}, \underset{m}{\mathbb{P}}\right) \leq T\mathtt{KL}\left(\mathtt{Bern}\left(\frac{1}{2} + \varepsilon\right), \mathtt{Bern}\left(\frac{1}{2}\right)\right) \leq 4T\varepsilon^2 \tag{30}$$

Plugging in Equations (29) and (30) in Equations (27) and (28) we get:

$$\underset{U}{\mathbb{E}}\left[T_{\leq -\delta}\right] \geq \underset{m}{\mathbb{E}}\left[T_{\leq -\delta}\right] - T\varepsilon\sqrt{2T}$$

and

$$\underset{L}{\mathbb{E}}\left[T_{\geq \delta}\right] \geq \underset{m}{\mathbb{E}}\left[T_{\geq \delta}\right] - T\varepsilon\sqrt{2T}$$

Finally, averaging Equations (25) and (26) and using the latter two Equations we get:

$$\max_{\nu \in \{U, L\}} \underset{\nu}{\mathbb{E}}\left[\mathcal{R}(T)\right) \geq \frac{\mathbb{E}_U\left[\mathcal{R}(T)\right] + \mathbb{E}_L\left[\mathcal{R}(T)\right]}{2} \geq \varepsilon\left(T - 2\varepsilon T\sqrt{2T}\right) \tag{31}$$

Tuning $\varepsilon = \frac{1}{3\sqrt{2T}}$ gives the result. ■

*Proof of Theorem 5.1.* We now assume without loss of generality that $2^\kappa = \lambda(\mathcal{A})/\lambda(\tilde{p}_\delta)$ for some constant $\kappa$. Let $\Phi$, such that $\Phi = \log\left(\frac{\lambda(\mathcal{A})}{\lambda(\tilde{p}_\delta)}\right)$, be a set of *phases* created from $\frac{T}{\Phi}$ consecutive rounds in $T$. We create the following problem instance described for clarity in 2 dimensions, and using truthful[15] agents.

First focus on phase $\phi = 1$ with feature vector $\mathbf{x}_t = \mathbf{u} = (1/2, 1/2)$, and specify the adversarial environments $U$ and $L$ exactly as in Lemma C.1. Then, after $T/\Phi$ rounds one of the two adversarial environments must have caused regret of at least $\sqrt{\frac{T}{162\Phi}}$ (Lemma C.1). If that environment was $U$, then it means that the majority of the labels for $\mathbf{u}$ is $+1$ and hence, the best-fixed action in hindsight is $\alpha_U^*$ such that $\langle\alpha_U^*, \mathbf{u}\rangle \geq 2\delta$. So fix the next phase's feature vector to be $\mathbf{x}_t = \mathbf{u} = (1/2, 5/8)$. Otherwise, if that environment was $L$, then it means that the majority of the labels for $\mathbf{u}$ is $-1$ and hence, the best-fixed action in hindsight is $\alpha_L^*$ such that $\langle\alpha_L^*, \mathbf{u}\rangle \geq 2\delta$ and you should fix the next feature vector to be $\mathbf{x}_t = \mathbf{u} = (1/2, 3/8)$. The reason for making this seemingly arbitrary choice is that we need to guarantee that one of the best-fixed actions for all previous phases, has survived and is still in the active set of actions in the current phase.

The general pattern that we follow is the following. At phase $\phi \in [\Phi]$, choose feature vector $\mathbf{x}_\phi = \left(\frac{1}{2}, \frac{1}{4}\left(1 + \kappa_\phi \cdot 2^\phi\right)\right)$, where $\kappa_\phi$ is a phase-specific constant defined as follows. If at phase $\phi$, the adversarial environment incurring Stackelberg regret $\sqrt{\frac{T}{162\Phi}}$ was environment $U$, then, $\kappa_{\phi+1} = 2\kappa_\phi + 1$, else $\kappa_{\phi+1} = 2\kappa_\phi - 1$. This is enough to establish that at *any* phase $\phi$, there exists an action that would have been the best-fixed for *all* previous phases despite which sequence of adversarial environments $U, L$ occurred. Another way to view this is similar to the polytope partitioning outlined in Figure 3; presenting these feature vectors, we can guarantee that the polytope that held the best-fixed action so far, is still active. As a result, the regret for all $\Phi$ phases is equal to the sum of regrets of each phase. Additionally, the Lebesgue measure of the smallest polytope is $\underline{p}_\delta(\phi)$ and is a non-increasing function of the phases, even if it is announced to the learner that $\delta = 0$. As a result,

$$\mathbb{E}\left[\sum_{t\in[T]} \ell(\alpha_t, \mathbf{r}_t(\alpha_t), y_t)\right] - \min_{\alpha^\star\in\mathcal{A}} \mathbb{E}\left[\sum_{t\in[T]} \ell(\alpha^\star, \mathbf{r}_t(\alpha^\star), y_t)\right] \geq \log\left(\frac{\lambda(\mathcal{A})}{\lambda(\tilde{p}_\delta)}\right)\frac{1}{9\sqrt{2}}\sqrt{\frac{T}{\log\left(\frac{\lambda(\mathcal{A})}{\lambda(\tilde{p}_\delta)}\right)}}$$

∎

# D    Appendix for Section 4.1

## D.1    Implementing GRINDER for Continuous Action Spaces

In order to implement GRINDER, we used the `polytope` library[16], which is part of the TuLiP python package. Other than some rounding-error fixes, we did not intervene with the core methods of the package.

In order to implement the 2-stage action draw method, we first chose a polytope (according to the probability function prescribed by GRINDER) and then, by using *rejection sampling* from the bounding box around the polytope, we chose the action associated with it. Note that this is equivalent to the theoretical 2-stage draw.

In order to speed up our algorithm's performance, we also used the heuristic of bounding the allowable volume of any polytope to be greater than or equal to 0.01, but in all the simulations that we tried, we saw comparable regret results even without the heuristic.

## D.2    Logistic Regression Oracle

In this subsection, we will outline our implementation of the logistic regression algorithm on the agents' past data, which serves as an estimate of the in-probability for each action. For ease of

Figure 6: GRINDER vs. EXP3 for utility function from Eq. (32). From left to right: discrete $\mathcal{A}$ (accurate and regression oracle), continuous $\mathcal{A}$ with $\delta = 0.05, 0.10, 0.15$ and continuous $\mathcal{A}$ with $\delta = 0.05, 0.3, 0.5$. Solid lines correspond to average regret/loss, and opaque bands correspond to 10th and 90th percentile.

exposition, we provide the description of the oracle for the case of a predefined action set, and subsequently, we outline the way it generalizes to the continuous implementation.

Before we embark on this, allow us first to observe that we already have a very crude (but potentially useful) lower bound for *every* action $j \in \mathcal{A}$. Indeed, each action *always* updates itself, and actions that belong in the upper and lower polytope sets are always updated by all actions within these sets. The latter is due to the fact that for any hyperplane chosen within these sets, there is no possible manipulation from the perspective of the agent. We denote this crude lower bound for each action $j \in \mathcal{A}$ by $c^j$.

Labels are defined as $l_i^j = 0$ if action $j$ was *not* updated at round $i$[17], and 1 otherwise. As a first step, this oracle computes for each action $j \in \mathcal{A}$ the probability that each action from $\mathcal{A}$ updates $j$, by using a logistic regression[18] with feature vectors the set $H_{1:t}$, and $L_{1:t}^j$ as the labels. Let $p_i^j, i \in \mathcal{A}$ correspond to the output probabilities, i.e., $p_i^j$ encodes the probability that action $j$ will be updated by action $i$. The in-probability of action $j$ is ultimately defined as:

$$\overset{in}{\mathbb{P}}[j] = \max \left\{ \sum_{i \in \mathcal{A}} p_i^j \pi_t[i], c^j \right\}$$

At a high-level, it is not hard to see how this can generalize to the continuous grinding case; instead of actions, one now uses whole *polytopes*. The implementation, however, becomes significantly messier, as we need to propagate the history of past data for each polytope to its grinded sub-polytopes.

### D.3  Different Utility Function and Distribution of Datapoints

The utility function that we assume for the agents at this subsection, is similar to the one studied by Dong et al. [17], specifically:

$$u_t(\alpha_t, \mathbf{r}_t(\alpha_t), y_t) = \delta \cdot \langle \alpha_t, \mathbf{r}_t(\alpha_t) \rangle - \|\mathbf{x}_t - \mathbf{r}_t(\alpha_t)\|_2 \tag{32}$$

for values of $\delta = 0.05, 0.10, 0.15, 0.3, 0.5$. Similarly to the paper's main body, we run GRINDER against EXP3 for a horizon $T = 1000$, where each round was repeated for 30 repetitions.

Fig. 6 presents the results for the case where the $+1$ labeled points are drawn as $\mathbf{x}_t \sim (\mathcal{N}(0.7, 0.3), \mathcal{N}(0.7, 0.3))$ and the $-1$ labeled points are drawn from $\mathbf{x}_t \sim (\mathcal{N}(0.4, 0.3), \mathcal{N}(0.4, 0.3))$. The performance of GRINDER compared to EXP3 is similar to the one that we saw in Sec. 4.1 for the case of the different utility function. GRINDER outperforms EXP3, and its performance degrades as the power of the agent (i.e., $\delta$) increases. We also see that in this case, the regression oracles are performing slightly worse that the regression oracles for the case of the utility function analyzed in Sec. 4.1.

Figure 7: GRINDER vs. EXP3 for "harder" distribution of labels. From left to right: discrete $\mathcal{A}$ (accurate and regression oracle), continuous $\mathcal{A}$ with $\delta = 0.05, 0.10, 0.15$ and continuous $\mathcal{A}$ with $\delta = 0.05, 0.3, 0.5$. Solid lines correspond to average regret/loss, and opaque bands correspond to 10th and 90th percentile.

Finally, in Fig. 7 we present the results of our simulations of running GRINDER against EXP3, when the agents' utility function is defined by Equation (32), and the distribution of labeled points is the following: the $+1$ labeled points are drawn as $\mathbf{x}_t \sim (\mathcal{N}(0.6, 0.4), \mathcal{N}(0.4, 0.6))$ and the $-1$ labeled points are drawn from $\mathbf{x}_t \sim (\mathcal{N}(0.4, 0.6), \mathcal{N}(0.6, 0.4))$. We note that while GRINDER still outperforms EXP3 its performance has become worse than what we saw in Fig. (6). This is due to the fact that in this new distribution of points creates much higher overlap of labels and there are fewer points for which a perfect linear classifier exists. This is also exhibited by the fact that EXP3's performance is getting better in the horizon of $T$ rounds compared to any single fixed action.

## Footnotes

[13]Quasilinearity in $\alpha$ establishes that $L_{f,g}$ which is used by Lemma A.6 will be linear in $\|\alpha' - \alpha\|$.

[14]We use the subscript $f, g$ in the Lipschitzness constant to denote the fact that it depends on the two functions $f$ and $g$.

[15]This way we establish the creation of the $\sigma$-t induced polytopes from the way that we construct the sequence of datapoints.

[16]https://github.com/tulip-control/polytope/tree/master/requirements

[17]In other words, action was at a distance less than $2\delta$ from the best-response of the round.

[18]Technically, we run a different logistic regression for every action in $\mathcal{A}$.