[Reviews · NeurIPS 2020]

Review 1

Summary and Contributions: The paper studies a repeated strategic classification setting where the goal is to achieve small Stackelberg regret (which is similar to external regret except that the opponent with limited power always best responds to the action played). The authors consider linear classifiers as actions and assume the opponent can respond to the classifier chosen at each time by moving its true features within a ball of radius \delta. The loss incurred is 1 if the label assigned by the classifier to the modified point is different from its true label, and 0 otherwise. The main result is an algorithm with regret \sqrt{T \log T * D} where D is a data-dependent quantity. The authors also show a lower bound that appears to (see below) nearly match the above regret bound.

Strengths: The authors consider a relatively novel notion of regret (the Stackelberg regret), which as they show, is inconsistent with the classical external regret, and better captures the strategic aspect in online learning. The regret bound appears (see below) nearly optimal. The experiments suggest that the proposed method outperforms EXP3 in practice.

Weaknesses: My main concern about the paper is that the main results (i.e., the regret upper and lower bounds) are presented in a somewhat vague way. For the upper bound (Theorem 4.1), some questions I had are the following: what if \mathcal{A} is a finite set? I guess then the ratio between the measures should be |\mathcal{A}|, but what if \mathcal{A} is a mixture of positive-measure polytopes and discrete points? I don't think the regret would be unbounded in that case, but what should the right bound look like? For the lower bound (Theorem 5.1), what is a \sigma_t-induced polytope? How is \tilde{p} different from \underline{p} (I guess there is some difference because otherwise the authors would just write \underline{p}...)? And how does this difference affect the extent to which the upper and lower bounds match? I'd appreciate it if the authors could further clarify the relation between the two bounds, especially because if they don't really match, then the data-dependent upper bound would appear much weaker. (And I will increase my score if the authors give a satisfactory explanation.)

Correctness: I did not check the proofs in appendices, but I'm willing to believe the main results are correct.

Clarity: The paper is overall well-written. At places the technical exposition could be improved -- for example, I can understand Algorithm 2 by going back and forth between the algorithm environment and the descriptive paragraphs, but the (abuse of) notation there confuses me a little (also see detailed comments) and some sentences just didn't make sense at first glance. When I came back to them I see what they mean, but still I feel there's some way to better present the algorithm which by iteself carries quite some intuition.

Relation to Prior Work: The discussion about related work is overall satisfactory, though I'd like to see more comparison with results from the online learning community (about which I don't know much). For example, what are the most closely related results there (I'm sure similar settings with the external regret have been considered somewhere), and why are they different from results in this paper?

Reproducibility: Yes

Additional Feedback: ===== after rebuttal ===== I am satisfied with the response, which in particular nicely addresses my main concern about the paper. I have raised my score and recommend acceptance. ==================== Definition of \mathcal{A} appears inconsistent -- initially \mathcal{A} \subseteq [-1, 1]^{d + 1} is a point set, but later in the algorithm it seems \mathcal{A} = \mathcal{P}_0 is treated as a collection of polytopes. In the experiments, for coninuous actions, it seems exp3 is executed on a nonadaptive discretization of the action space. Can this be interpreted as illustrating the advantage of adaptive discretization over nonadaptive ones? Line 96: this is not wlog but is actually a requirement on u_t, right? Line 103: this doesn't seem to match Eq (1) Line 188: comma => full stop? Line 289: what does this mean: agents are truthful but the learner doesn't know that? In principle one can always consider algorithms which "behave like they know that" right? If that's the case, then is this equivalent to a lower bound in the classical bandit setting (with external regret)?


Review 2

Summary and Contributions: The authors study a learning setting whereby agents report features in an strategic online fashion. To be more specific, the authors study an online learning setting over T rounds, where a learner must provide a linear classifier at each round. Nature provides a feature vector to be classified for a given round, and a strategic adversary takes this vector and perturbs it optimally within a bounded \delta neighbourhood in the \ell_2 norm, while maintaining the original label of the feature vector. These perturbations can be beneficial to a strategic adversary if the relevant feature vector is close to the decision boundary of the linear classifier set by the learner. As a motivating example, the authors mention that spam mail can strategically perturb its contents (while still remaining spam) to bypass a classifier. In this setting, the authors measure the performance of an online classifier with two notions: external regret and Stackelberg regret. For external regret, the learner fixes the strategic best responses to his committed classifiers over the time horizon, and in hindsight picks the single best cumulative action against these strategic reports. For Stackelberg regret, the learner simply chooses the best action over the time horizon, while permitting the strategic adversary to best respond to this commitment at each turn. The authors begin by showing that these two notions of regret are incompatible in the worst case. There are scenarios where one is sublinear whereas the other is linear and vice versa. Subsequently, the authors give an algorithm for achieving sublinear Stackelberg regret in this setting, as well as a matching lower bound. The algorithm is based off of a geometric intuition on polytopal partitions of leader action spaces.

Strengths: The model is very natural, and the results use interesting mathematical techniques to produce strong results. Given the increased interest in incentive-aware learning mechanisms, the material is highly relevant to NeurIPS. The authors have also addressed reviewer concerns in the rebuttal phase well. My score remains the same.

Weaknesses: Overall I can think of few weaknesses along the theoretical axes mentioned above. It would be interesting to see more extensive simulation results, as well as graphical comparison with external regret in the simulation setting to see how incompatible these objectives are in practice.

Correctness: The methodology seems correct to me, though I have some notational questions which are mentioned in the clarity section.

Clarity: Overall the paper is well-written, but there is one confusing unexplained notation and minor changes I might recommend. Confusing notation: In Algorithm 2 (Grinder), the loss function is extended to polytopes and adversarial responses to a learner action polytope. I'm guessing this means the worst case loss over actions in the polytope and corresponding strategic tweaks to x_t, but this is unclear as is in the text. Small changes I might recommend: -In Figure 1, it would be useful to know how h' and h classify each of their relevant halfspaces. It can be inferred from the material, but it would be nice to know off the bat. -The proof sketch of theorem 3.3. (on page 4) also mentions x^2,x^2,x^3,x^4. I think the first was meant to be x^1 -

Relation to Prior Work: This is handled well.

Reproducibility: Yes

Additional Feedback:


Review 3

Summary and Contributions: The authors introduce a game between a learner and (a stream of) agents. In every round, the learner commits to a linear classifier, and an agent, equipped with a feature vector and label, reports a modified version of his feature vector. The learner then suffers a binary loss for incorrect classification, and the agent receives a utility according to his underlying distribution, which is unknown to the learner. The authors assume that an agent can misreport in a ball of a constant radius (the radius is known to the learner), centered around his actual feature vector. The main technical contribution of the paper is Grinder, an algorithm that achieves sublinear Stackelberg regret. Informally, starting from a set of available actions, it interacts with agents and “grinds” the action space (polytopes, every point in which is a linear classifier) further after incurring a loss (or rather, observing the label). The critical observation is that all actions (classifiers) that classify the reported feature vector as +1 (similarly for -1) and do not intersect the 2*delta ball centered at the reported feature vector would have incurred the same loss w.r.t. the actual, unobserved feature vector. The authors bound the Stackelberg regret of their algorithm, and later present a nearly-matching lower bound. Additionally, they provide simulations on synthetic data to demonstrate their setting and evaluate their algorithm.

Strengths: • The topic is timely and valuable. This work departs from classical adversarial multi-armed bandits (MAB) in that the main objective is the Stackelberg regret and not the external regret. The authors also shed light on the incompatibility between these two notions of regret terms. Importantly, in this paper, agents do not wish to maximize the learner’s loss but rather their utility. While a similar offline model [13] (citation numbers are as in the paper), as well as a convex online model [17], were suggested recently, the model proposed in this paper offers a different perspective and proposes a genuine technical challenge. • While Grinder, the main algorithmic contribution looks natural, proving its guarantees seems non-trivial to me. As far as I can tell, the proofs are sound.

Weaknesses: • The protocol the authors use looks a bit artificial. Why is this the order of play? What is the motivation for the agent observing the classifier? Why can Nature pick the feature vector arbitrarily (and not stochastically, from an unknown but fixed distribution)? Why can’t the learner commit to a distribution over classifiers and not a single classifier? I’m not claiming against the proposed model, but I wish the authors could better justify their modeling decisions. • As the authors themselves admit, Grinder is impractical due to runtime complexity blowup. It also means that the experimental part is extremely synthetic. On the positive side, the authors acknowledge this weakness and provide several options for reducing the algorithm’s runtime (approximation oracle and an ML-based approach with Logistic Regression). • In many ML applications, we use convex surrogates to non-convex problems (to illustrate, finding the best linear predictor in the offline problem w.r.t. binary loss takes exponential time in the dimension). These surrogates can be optimized efficiently AND turn to be robust w.r.t. binary loss (e.g., SVM or Logistic Regression). This raises two questions: 1) Are there any convex surrogate that could fit this setting? Even if their formal guarantees are weaker or they are trivial to implement, and 2) In the simulations part, the authors do not consider a convex surrogate-based algorithm as a baseline, which I think is misleading. Sure, we expect their algorithm to work better, but by how much? Quantifying this extent could help to assess the tradeoff between runtime and accuracy.

Correctness: The claims, methods, and the empirical methodology look correct

Clarity: The paper is generally well-written, despite several points the authors could improve (see below).

Relation to Prior Work: The authors include a comprehensive discussion.

Reproducibility: Yes

Additional Feedback: Overall, I think that this paper fits the NeurIPS standard and will be interesting to the community, so I recommend accepting it. However, I have a few questions I wish the authors could answer: • The issue with {\hat y} is puzzling. Almost all prior work on strategic classification assumes that the agents can only misreport their feature vectors while the labels are the agents’ inherent property. In this paper, as a byproduct of defining h*, agents can hypothetically change their label too. However, the semi-formal discussion in lines 96-100 forbids it. Formally, this is equivalent to restricting the agent (the optimization problem in line 92) to pick z that with h*(z)=y_t. It is already assumed implicitly, so why not say this explicitly? Am I missing something here? • Relating to the previous point, how would the theoretical results change if an ideal classifier h* does not exist? (two feature vectors can get different labels) • A crucial component of the modeling is delta, the misreporting radius. The authors hide the dependence on delta from the bound in Theorem 4.1, which may be reasonable. However, as far as I can tell, it is also hidden in the proof of this statement. What is the dependence on delta? If there is no natural way of expressing it, the author could say it. • The example of the utility function in line 102: the authors probably missed the “sign” function inside the indicator in the right-hand-side. More profoundly, the text that follows the equation contradicts the formula: it says that the agent gains delta’ if his data point is labeled +1. However, the indicator activates for correct classification (classified as {\hat y_t}), even if both the actual label and predicted one are -1. Am I missing something? • About the convex surrogate: I wish the authors could address my statement about the convex loss. Can they augment their simulations with an appropriate convex surrogate baseline? What is that baseline? Additional comments: • (a matter of personal taste) Why did the authors separate the related work (line 66) from the model comparison (105)? IMO, it disrupts the flow of the paper. • Line 1 of the algorithm is impossible to parse. The authors should consider breaking it into several lines. • In Figure 4, what’s the difference between the middle and right figures? Is it just different values of delta with the same setup? • Typo in line 788 (in the appendix): “the the” ________________________________________________ Post-rebuttal feedback: The authors address my main concerns, and provide satisfying answers; therefore, my overall score remains positive.

[Author Response · NeurIPS 2020]

We *deeply* thank the reviewers for their very thoughtful reviews that will help improve our paper. All of them commended
the novelty of our model and the results. We address them separately below. The citation numbers are from the paper.

**Lines 102 – 103 (R1, R4):** We are indebted to **R1** and **R4** for highlighting this typo. The utility function that matches
the (correct) text is equal to: $\delta' \cdot \mathbb{1}\{\mathrm{sgn}(\langle \alpha_t, \mathbf{r}_t(\alpha_t, \sigma_t)\rangle) > 0\} - \|\mathbf{x}_t - \mathbf{r}_t(\alpha_t, \sigma_t)\|_2$. We will fix it in the revision.

**R1: (Upper Bound – Thm 4.1.)** If $\mathcal{A}$ is a finite set, then GRINDER becomes similar to standard[1] feedback graph (FG)
algorithms ([1]), but the FG is built according to $\delta$-BMR agents, and its regret scales as $O(\sqrt{\alpha(G)T\log T})$ ($\alpha(G)$ being
the independence number of the FG). We will add a clarifying paragraph after line 271. The case where $\mathcal{A}$ includes both
continuous and discrete sets of actions is a great direction for future work; in the standard online learning setting, results
only for metric spaces are known, so it is a big undertaking for cases where $\ell(\alpha, \mathbf{r}_t(\alpha))$ is not Lipschitz (lines 72 – 73).

**R1**: **(Lower Bound)** The $\sigma_t$-induced polytope sets are defined in lines 228 – 230, but following **R1**'s comments we
will move the definition outside of the proof. At a high level, these polytopes can be constructed in the dual space by a
super powerful learner who knows $\{\sigma_t\}_{t\in[T]}$ (hence the name), but *not exactly the agents' utility function* (footnote
10). These polytopes (with $\tilde{p}$ being the smallest among them) are not affected *or observed* by GRINDER. We use
them only as tools for the proof of Thm 4.1. and the lower bound. GRINDER's polytopes (lines 200 – 214) differ
from $\sigma_t$-induced ones in $\mathrm{poly}(d)$[2] factors, which translates to a $\log(\mathrm{poly}(d))$ gap between the two types due to the
logarithmic dependence of the regret on the smallest polytope (Thm 4.1.). So our bounds are matching up to logarithmic
factors (hence the "nearly"). **R1** was rightly confused, and we will add a clarifying sentence.

If the learner *knew* that the *agents are truthful*, then we would have a standard bandit problem. However, she does *not*
know neither $\mathbf{x}_t$, nor the agents' precise utility function (hence the dependence on the $\sigma_t$-induced polytopes and $\tilde{p}$).

**R1**: **(Related Work)** The strategic nature of our agents (making $\ell(\alpha, \mathbf{r}_t(\alpha))$ not Lipschitz) is what differentiates us
from any related standard online learning setting (lines 70 – 73). We will include an extra sentence highlighting this.

**R1**: **(Additional Feedback)** $\mathcal{P}_0$ should be $\mathcal{P}_0 = \{p(\mathcal{A})\}$ ($p(\mathcal{A})$ being the polytope representation of $\mathcal{A}$). **R1** is also
right that GRINDER illustrates an advantage of adaptive discretization over non-adaptive ones[3]. In line 96, this is true
for the settings of interest (e.g., spam identification). We will clarify accordingly lines 196 and 96 respectively.

**R2**: We thank **R2** for the positive comments, the typos, the suggestion for Fig. 1, and $\ell(p, \mathbf{r}_t(p))$ (lines 210 – 213) all of
which we will incorporate. We have extra simulations in Fig. 5, 6 of the paper, and we will also include the benchmark
that was suggested by **R4** (Fig. 1). For the incompatibility, we remark that it is a worst-case result and that EXP3 (Fig. 4,
5, 6) could also be viewed an algorithm that minimizes *external* regret.

**R4: (Model)** The schools' admission process can be thought of as another example of our protocol: the school commits
to a classifier, students get to "observe" it via proxies (e.g., past admitted students), and then, according to their
original point (e.g., features of their resume) and their manipulation power (e.g., money they spend on SAT) they can
best-respond. The distribution over classifiers is an excellent direction for future work, as it can "enforce" exploration
over the dataset. The modeling of nature choosing $\mathbf{x}_t$'s adversarially is because we wanted to take a worst-case approach,
but we conjecture that having the stochasticity that **R4** mentions would lead to better regret guarantees, and it is an
exciting avenue for future work. We agree with **R4** regarding $h^{\star}$[4]. We will add these clarifications to our model.

**R4**: **(Dependence on $\delta$)** $\delta$ affects the size of $\underline{p}$ via the definition of the $\sigma_t$-induced polytopes (lines 228 – 230), and the
analysis of $Q_1$ (Eq. (2), lines 235 – 243). We will change $\underline{p}$ to $\underline{p}(\delta)$ to make this clearer.

**R4: ("Convex" Surrogates)** *No standard convex surrogate can be used* since the learner does
not know precisely *function* $\mathbf{r}_t(\alpha)$. Hence, the learner cannot guarantee that $\ell(\alpha, \mathbf{r}_t(\alpha))$ is *convex*
in $\alpha$, even if $\ell(\alpha, \mathbf{z})$ ($\mathbf{z}$ being independent of $\alpha$) is convex in $\alpha$! Concretely, we will include
this counterexample: let $h = (1, 1, -1), h' = (0.5, -1, 0.25)$ two hyperplanes, a point $\mathbf{x} =$
$(0.55, 0.4), y = +1, \delta = 0.1$, and let $\ell(h, \mathbf{r}(h)) = \max\{0, 1 - y \cdot \langle h, \mathbf{r}(h)\rangle\}$ (i.e., hinge). We
will show that when $(\mathbf{x}, y)$ is a $\delta$-BMR agent, $\ell(\alpha, \mathbf{r}(\alpha))$ is no longer convex in $\alpha$. Take $b = 0.5$

Figure 1

and construct $h_b = 0.5h + 0.5h' = (0.75, 0, -0.375) = (1, 0, -0.5)$. $(\mathbf{x}, y)$ only misreports
to (say) $(0.61, 0.4)$ when presented with $h$ (as $h_b$ and $h'$ classify $\mathbf{x}$ as $+1$). Computing the loss: $\ell(h_b, \mathbf{r}(h_b)) =$
$0.95, \ell(h, \mathbf{r}(h)) = 0.99$ and $\ell(h', \mathbf{r}(h')) = 0.875$, so, $\ell(h_b, \mathbf{r}(h_b)) > b\ell(h, \mathbf{r}(h)) + (1 - b)\ell(h', \mathbf{r}(h'))$. Since in
general $\ell(\alpha, \mathbf{r}(\alpha))$ is not convex, it is unfair to compare Bandit Gradient Descent (BGD) with GRINDER (so we did
not include these experiments originally). Since **R4** asked for them, we will include Fig. 1, where GRINDER greatly
outperforms BGD. Identifying surrogate losses that are convex against $\delta$-BMR agents is an exciting future direction!

## Footnotes

[1]This is *precisely* how we implemented GRINDER for our simulations with the discrete set of actions (left graphs in Fig. 4, 5, 6)!

[2]$\sigma_t$ polytopes are defined by hyperplanes with a margin of $2\delta$, but GRINDER's polytopes are defined with a margin of $4\sqrt{d}\delta$.

[3]But we cannot use the "standard" adaptive discretization techniques ([21,9]) since $\ell(\alpha, \mathbf{r}_t(\alpha))$ is not Lipschitz (lines 72 – 73).

[4]We introduced $h^{\star}$ in order to explain that the mapping $\mathcal{X} \to \{-1, +1\}$ does not have to be linear, but it can be arbitrary.


[Meta-Review · NeurIPS 2020]

The author response addressed all major concerns of the reviewers. The paper makes an interesting and important contribution to strategic classification, and we are happy to recommend acceptance.